# The Emergence of Abstract Thought in Large Language Models Beyond Any Language

Yuxin Chen[1]*    Yiran Zhao[2]*    Yang Zhang[3]    An Zhang[1]    Kenji Kawaguchi[1]    Shafiq Joty[2]
Junnan Li[2]    Tat-Seng Chua[1]    Michael Qizhe Shieh[1]†    Wenxuan Zhang[4]†

[1] National University of Singapore    [2] Salesforce AI Research    [3] Peking University
[4] Singapore University of Technology and Design

## Abstract

As large language models (LLMs) continue to advance, their capacity to function effectively across a diverse range of languages has shown marked improvement. Preliminary studies observe that the hidden activations of LLMs often resemble English, even when responding to non-English prompts. This has led to the widespread assumption that LLMs may "think" in English. However, more recent results showing strong multilingual performance, even surpassing English performance on specific tasks in other languages, challenge this view. In this work, we find that LLMs progressively develop a core language-agnostic parameter space—a remarkably small subset of parameters whose deactivation results in significant performance degradation across all languages. This compact yet critical set of parameters underlies the model's ability to generalize beyond individual languages, supporting the emergence of abstract thought that is not tied to any specific linguistic system. Specifically, we identify language-related neurons—those are consistently activated during the processing of particular languages, and categorize them as either shared (active across multiple languages) or exclusive (specific to one). As LLMs undergo continued development over time, we observe a marked increase in both the proportion and functional importance of shared neurons, while exclusive neurons progressively diminish in influence. These shared neurons constitute the backbone of the core language-agnostic parameter space, supporting the emergence of abstract thought. Motivated by these insights, we propose neuron-specific training strategies tailored to LLMs' language-agnostic levels at different development stages. Experiments across diverse LLM families support our approach.[1]

## 1 Introduction

As large language models (LLMs) continue to advance (OpenAI, 2023; Touvron et al., 2023a; Hurst et al., 2024; Yang et al., 2024a; Team et al., 2024), their performance across a wide range of languages (known as multilingual capability) has markedly improved over the past years (Le Scao et al., 2023; Yang et al., 2024b; Üstün et al., 2024). Despite this progress, several studies have observed that LLMs tend to "think in English", often using it as an internal language of thought even when processing inputs in other languages (Wendler et al., 2024; Zhao et al., 2024b; Schut et al., 2025a). This phenomenon has led to the hypothesis that LLM performance in non-English languages is inherently constrained by their capabilities in English (Qin et al., 2023; Liu et al., 2024).

---

*Equal Contribution.

†Corresponding Authors.

[1]Our codes are available at `https://github.com/chenyuxin1999/Abstract_Thought`.

Yet, more recent findings complicate this narrative: some studies found that LLMs can actually outperform their English-language performance on certain tasks in other languages (Zhao et al., 2025c; Gemma Team et al., 2024b, 2025), indicating that non-English processing may not always rely on English as an intermediate language. These conflicting observations raise a deeper research question: *Do LLMs think in the distinct space of each language*, or *Do they operate in a higher-level language-agnostic space beyond any specific language?* In other words, whether the trend of non-English performance compared to English indicates **the emergence of abstract thought within LLMs?**

In this work, we explore the existence and development of abstract thought in LLMs by analyzing how individual neurons, responsible for models' thinking, respond to multilingual queries. Here each neuron corresponds to a row or column in the model's parameter matrices and is considered activated if its removal significantly alters the model's output (Frankle and Carbin, 2018; Tang et al., 2024; Wang et al., 2025a). We begin by identifying neurons activated when the model processes inputs in specific languages, referred to as *Language-Related Neurons*. To investigate whether these neurons become increasingly specialized for specific languages or potentially exhibit more general and language-agnostic functionality, we distinguish between *Language-Exclusive Neurons*, which are activated only for one language, and *Language-Shared Neurons*, which are consistently activated across all languages considered. Figure 1 (top) shows a positive correlation between multilingual ability and the proportion of language-shared neurons across different generations of LLMs. This suggests that **as multilingual performance improves, a greater proportion of language-related neurons are shared across languages.**

Building on the observed positive relationship between language-shared neurons and multilingual capability, as well as the finding that LLMs increasingly outperform in non-English languages on certain tasks, we hypothesize that shared neurons may gradually assume more fundamental roles beyond merely supporting multilingual processing. Accordingly, rather than focusing solely on how the proportion of language-

Figure 1: **(Top)** The trendline of shared neuron proportion rises with model release date (see Section 2.2). **(Bottom)** The trendline of shared neuron importance also grows, indicating their increasing language-agnostic property (see Section 2.3).

shared neurons evolves across model generations, it is essential to evaluate their functional significance relative to language-exclusive neurons. If shared neurons contribute more critically to multilingual processing than language-exclusive neurons—which also participate in language tasks and should be comparably important in principle—this would indicate that shared neurons have evolved into *Language-Agnostic Neurons*, which go beyond shared activation patterns to support abstract functions like semantic reasoning and generalization. As these neurons evolve, they support increasingly abstract thought that transcends linguistic boundaries. As shown in Figure 1 (bottom), language-shared neurons exhibit a markedly growing importance in multilingual processing relative to language-exclusive neurons, signaling **the emergence of language-agnostic properties and potentially, the development of abstract thought in LLMs.**

Inspired by the insights discussed above, we propose a set of targeted neuron training strategies aimed at enhancing the multilingual capabilities of LLMs. These methods are tailored based on the presence or absence of language-agnostic neurons, which serve as an indicator of the emergence of abstract thought within the model. For LLMs that lack language-agnostic neurons, the model is likely still under-trained; thus, training any language-related neurons can contribute to improving multilingual performance. In contrast, in LLMs where abstract thought has emerged, language-shared neurons have evolved into language-agnostic ones. As these neurons have reached a form of generalization,

further improvement through additional training is limited. In such cases, enhancing multilingual capabilities requires focusing on training language-exclusive neurons to better support language-specific nuances. We validate our approach through comprehensive experiments across diverse model series and release time. The results demonstrate that our training method, guided by the presence of language-agnostic properties, effectively enhances multilingual performance.

## 2 Metrics for Exploring Abstract Thought

In this section, we identify neurons associated with language processing, referred to as *Language-Related Neurons*, and, based on them, define several metrics to quantify and analyze the emergence of abstract thought in LLMs.

### 2.1 Language-Related Neurons

We identify language-related neurons as those that are consistently activated when processing inputs in a particular language, where a neuron is defined as a single row or column within the model's parameter matrices. Building on prior work in identifying important neurons in neural networks (Frankle and Carbin, 2018; Ni et al., 2023; Tang et al., 2024; Zhao et al., 2024b), we consider a neuron to be activated if its removal leads to a significant change in the resulting embedding. Formally, given an input sequence $x$ in a specific language, a neuron $\mathcal{N}$ is considered activated if

$$\|\mathcal{LLM}(x) - \mathcal{LLM}_{\ominus\mathcal{N}}(x)\|_2 \geq \sigma, \tag{1}$$

where $\mathcal{LLM}(x)$ denotes the output embedding when processing $x$, and $\mathcal{LLM}_{\ominus\mathcal{N}}(x)$ denotes the output when neuron $\mathcal{N}$ is deactivated, i.e., its parameters are set to zero. The threshold $\sigma$ specifies the minimum magnitude of change required to consider a neuron activated.

Furthermore, language-related neurons $\mathcal{N}_{\text{lang}}^{\ell}$ for a specific language $\ell$ are identified through

$$\mathcal{N}_{\text{lang}}^{\ell} := \left\{ \mathcal{N} \in \mathcal{LLM} \;\middle|\; \|\mathcal{LLM}(x) - \mathcal{LLM}_{\ominus\mathcal{N}}(x)\|_2 \geq \sigma, \; \forall x \in \ell \right\}. \tag{2}$$

Since sequentially deactivating neurons in Equation 2 is computationally expensive, we employ the parallel neuron detection methods proposed in Zhao et al. (2024b); Wang et al. (2025a). Further implementation details are provided in Appendix A. Details of how the activation threshold $\sigma$ is selected and validated are provided in Appendix B.

### 2.2 Language-Shared and Language-Exclusive Neurons

To investigate whether neurons become increasingly specialized for specific languages or exhibit language-agnostic behavior, we conduct a preliminary analysis of the proportion of *Language-Shared Neurons*, defined as language-related neurons that are consistently activated across all languages considered, and *Language-Exclusive Neurons*, defined as language-related neurons that are uniquely activated for individual languages and not shared across all languages. Formally, language-shared and language-exclusive neurons are defined as follows:

$$\mathcal{N}_{\text{shared}} := \bigcap_{\ell \in \mathcal{L}} \mathcal{N}_{\text{lang}}^{\ell}, \quad \text{and} \quad \mathcal{N}_{\text{exclusive}}^{\ell} := \mathcal{N}_{\text{lang}}^{\ell} \setminus \mathcal{N}_{\text{shared}}, \tag{3}$$

where $\mathcal{L}$ denotes the set of all languages under consideration. In other word, $\mathcal{N}_{\text{shared}}$ consistently exhibit high importance across inputs from different languages, while $\mathcal{N}_{\text{exclusive}}^{\ell}$ is the set of neurons specific to that language but not part of the shared set. Furthermore, we examine the proportion of language-shared neurons relative to language-exclusive neurons, defined as

$$\text{Shared Neuron Ratio} := \frac{|\mathcal{N}_{\text{shared}}|}{\frac{1}{|\mathcal{L}|} \sum_{\ell \in \mathcal{L}} |\mathcal{N}_{\text{exclusive}}^{\ell}|}, \tag{4}$$

which quantifies the extent to which individual neurons are shared across all languages as opposed to being specialized for specific ones. A higher ratio indicates a greater number of neurons that are commonly activated across languages, while a lower ratio suggests that more neurons are uniquely responsive to individual languages. A more detailed analysis of the shared and exclusive neurons, including their layer-wise and component-level distributions, can be found in Appendix C.

## 2.3 Language-Agnostic Neurons

As LLMs continue to improve in their ability to handle multiple languages, and even outperform their English capabilities on specific tasks, we hypothesize that shared neurons may gradually serve more fundamental functions beyond the processing of multiple languages. Consequently, rather than solely examining the evolution of the proportion of language-shared neurons across successive model generations, it is also crucial to assess their relative functional significance in comparison to language-exclusive neurons. Specifically, if language-shared neurons contribute significantly more to multilingual processing than language-exclusive neurons, this indicates a functional difference between the two, since both types are involved in language-related tasks and identified using the same criteria, they should be equally important if their roles are analogous. The discrepancy suggests that shared neurons have evolved into *Language-Agnostic Neurons*. Note that while language-shared neurons are activated across multiple languages, language-agnostic neurons reflect a higher level of abstraction. Rather than encoding language-specific features, they are hypothesized to support cognitive functions that transcend individual languages, such as semantic abstraction, reasoning, and generalization.

To investigate whether language-agnostic property emerge in language-shared neurons, we introduce the metric *Language-Shared Neuron Importance*, which quantifies the impact of deactivating language-shared neurons versus language-exclusive neurons on the model's performance in a given language. This is operationalized by measuring the change in perplexity ($\Delta\text{PPL}$) when each neuron group is ablated. A disproportionately larger increase in perplexity upon deactivating shared neurons would suggest their greater functional importance. Formally, we define the language-shared neuron importance for a language $\ell$ as:

$$\text{Imp}^\ell := \frac{\Delta\text{PPL}^\ell_{\text{shared}}/|\mathcal{N}_{\text{shared}}|}{\Delta\text{PPL}^\ell_{\text{exclusive}}/|\mathcal{N}^\ell_{\text{exclusive}}|}, \tag{5}$$

where $\Delta\text{PPL}^\ell_{\text{shared}}$ and $\Delta\text{PPL}^\ell_{\text{exclusive}}$ denote the changes in perplexity for language $\ell$ when shared and corresponding exclusive neurons are deactivated, respectively, and $|\mathcal{N}_{\text{shared}}|$ and $|\mathcal{N}^\ell_{\text{exclusive}}|$ represent the number of neurons in each group. A higher value of $\text{Imp}^\ell$ indicates that shared neurons contribute more significantly than language-exclusive neurons, thereby providing evidence for their language-agnostic role, since both types of neurons should exhibit comparable importance if their functions were equivalent.

To obtain an overall model-level estimation reflecting this trend across different languages, we compute the average importance across all languages and apply a logarithmic transformation to mitigate scale sensitivity. We refer to the resulting quantity as the *Language Agnostic Score*:

$$\text{Language Agnostic Score} := \log\left(1 + \frac{1}{|\mathcal{L}|}\sum_{\ell\in\mathcal{L}}\text{Imp}^\ell\right), \tag{6}$$

which quantifies the average degree to which language-shared neurons contribute in a language-agnostic manner across the evaluated languages. In contrast to the shared neuron ratio defined in Equation 4, which solely quantifies the number of shared neurons, the language-agnostic score incorporates the functional importance of neurons. Higher values suggest not only stronger language-agnostic behavior but also hint at the emergence of abstract thought in LLMs.

# 3 Emergence of Abstract Thought

## 3.1 Experiment Setup

**Evaluated Models** To comprehensively evaluate the emergence of abstract thought in LLMs throughout their development, we examine 20 open-source models encompassing diverse model families, release periods, and sizes. Specifically, we evaluate Llama series including LLaMA1-7B (Touvron et al., 2023a), Llama2-7B (Touvron et al., 2023b), Llama3.2-1B, Llama3.2-3B, Llama3-8B, Llama3.1-8B (Grattafiori et al., 2024), Qwen1.5-0.5B, Qwen1.5-1.8B, Qwen1.5-4B, Qwen1.5-7B (Bai et al., 2023), Qwen2-0.5B, Qwen2-1.5B, Qwen2-7B (Yang et al., 2024a), Qwen2.5-0.5B, Qwen2.5-1.5B, Qwen2.5-3B, Qwen2.5-7B (Yang et al., 2024b), Gemma-7B (Gemma Team et al., 2024a), Gemma2-9B (Gemma Team et al., 2024b), Gemma3-4B (Kamath et al., 2025).

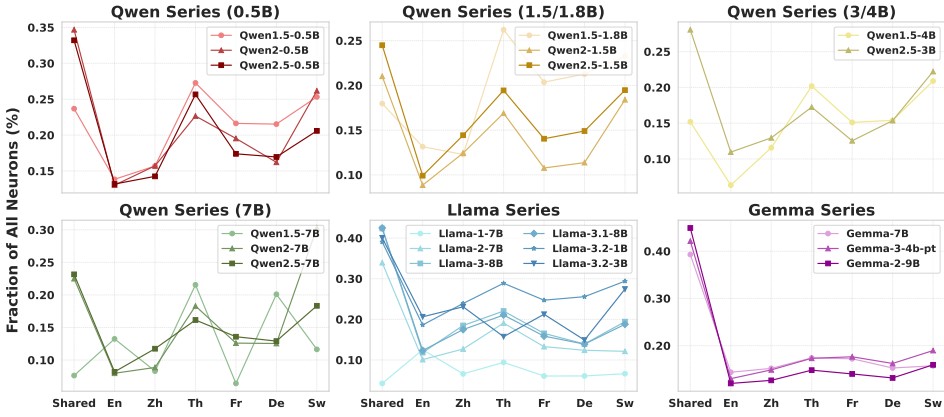

Figure 2: Neuron distribution across language-shared and language-exclusive neurons for six languages (En, Zh, Th, Fr, Dr, Sw) in various model series and scales. For each model, we present the fraction of shared neurons and exclusive neurons to the total number of neurons.

**Multilingual Benchmark**   We evaluate models across six typologically and resource-diverse languages: Chinese (Zh), English (En), Thai (Th), Swahili (Sw), French (Fr), and German (De). This selection spans high-resource, medium-resource, and low-resource languages, enabling a representative analysis of language-related neuron behaviors. For our analysis, we utilize the Multilingual Massive Multitask Language Understanding (MMMLU) dataset (OpenAI, 2024), a human-translated extension of the original MMLU benchmark (Hendrycks et al., 2021), available in 14 languages. In addition, we incorporate the Multilingual Grade School Math (MGSM) dataset (Shi et al., 2022), a translated version of GSM8K (Cobbe et al., 2021), which covers 10 languages. Together, these datasets provide quantitative measures of the models' multilingual capabilities.

**Neuron Detection Corpus**   For each language, we identify language-related neurons by analyzing activation patterns on 1000 sentences sampled from the OSCAR corpus (Abadji et al., 2022). To quantify the functional contribution of these neurons, we further compute perplexity changes caused by deactivating them, using the same language-specific OSCAR data. This unified framework allows us to assess both the proportion and the importance of language-specific and shared neurons across languages and model generations. More detailed illustration can be found in Appendix D.

## 3.2   Analysis on Shared Neuron Ratio

**Language-related neurons account for only a small proportion in LLMs.**   To develop a preliminary understanding of language-shared and language-exclusive neurons, we begin by analyzing the distribution of shared and language-exclusive parameters across all neurons within the model. For each language, we compute the proportion of language-shared and language-exclusive neurons relative to the total number of neurons in the model. Specifically, we calculate the ratios $\mathcal{N}_{\text{shared}}/|\mathcal{LLM}|$ and $\mathcal{N}_{\text{exclusive}}^{\ell}/|\mathcal{LLM}|$, where $\ell$ denotes a specific language. The results, illustrated in Figure 2, encompass six languages across multiple model series. It shows that only a small fraction of neurons, often fewer than $1\%$, play a critical role in processing language, underscoring the sparsity and selectivity of language-relevant neural activations. Furthermore, the quantities of language-shared and language-exclusive neurons are of similar magnitude, each coarsely estimated at around $0.3\%$ of the total number of neurons in the LLM.

To further explore the evolution of shared and exclusive neurons across models and over time, we compute the overall shared neuron ratio for each model, as defined in Equation 4, relate it to multilingual performance measured by MMMLU and MGSM, and present the results in Figure 3.

**The proportion of shared neurons increases with model evolution.**   We first group models from the same series and with similar parameter scales, as indicated by the shaded color regions in Figure 3. Within each group (e.g., Qwen1.5-7B, Qwen2-7B, and Qwen2.5-7B), we observe a steady and consistent increase in the shared-to-exclusive neuron ratio across generations. This growth closely parallels improvements in the model's multilingual ability, with an average Pearson correlation coefficient of $R = 0.92$ and a Spearman rank correlation of $\rho = 0.88$, indicating a strong and reliable

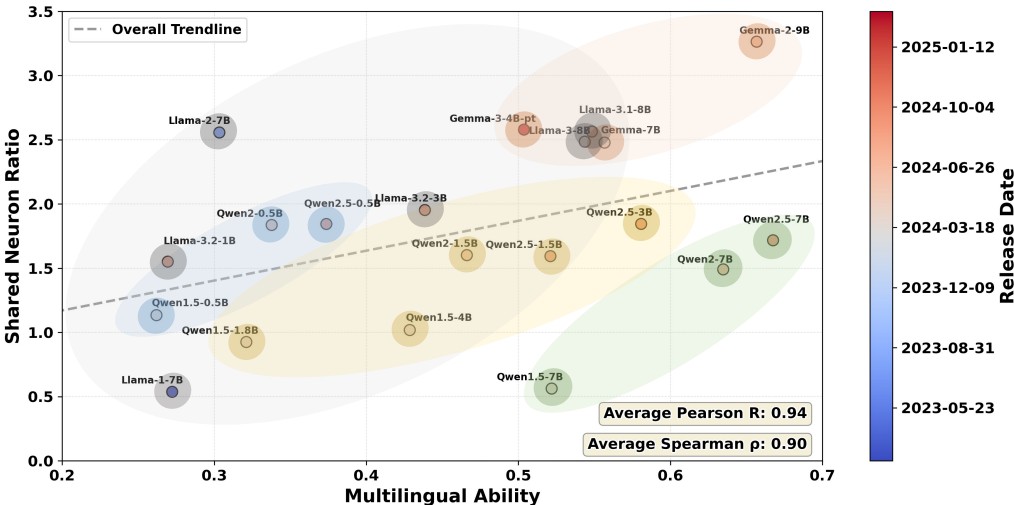

Figure 3: The relationship between multilingual ability and shared neuron ratio (as defined in Equation 4) across various models. Each point represents a model, color-coded by its release date. Shaded regions indicate groups of models within the same series and of comparable scale. The gray dashed line (- - - -) illustrates the overall trend: as models evolve, those with greater multilingual capabilities tend to exhibit a higher proportion of shared neurons.

relationship. In other words, later generations within the same series show a strong trend toward engaging more shared neurons for processing different languages.

**The increase of shared neuron proportion generalizes across model families.** Beyond individual model series, we observe that the positive relationship between the proportion of shared neurons and multilingual capability generally persists across different model families, as illustrated by the gray dashed line (- - - -) in Figure 3. Despite differences in architecture design and pretraining corpora, models with stronger multilingual ability tend to activate a larger proportion of shared neurons. For instance, the Gemma series exhibits both the most strong multilingual performance and the highest shared-to-exclusive neuron ratio. This consistency across diverse architectures suggests that progressively leveraging shared neurons may be a general strategy adopted by multilingual LLMs, regardless of their specific design choices.

### 3.3 Analysis on Language Agnostic Score

The above observations raise a further question: whether the shared neurons not only occupy a larger proportion of the language-related neuron set, but also *contribute more critically to multilingual processing*, effectively functioning as language-agnostic neurons. To address this question, we investigate how the language-shared neurons importance, i.e., language agnostic score defined in Equation 6, evolves alongside multilingual capability across different generations of large language models.

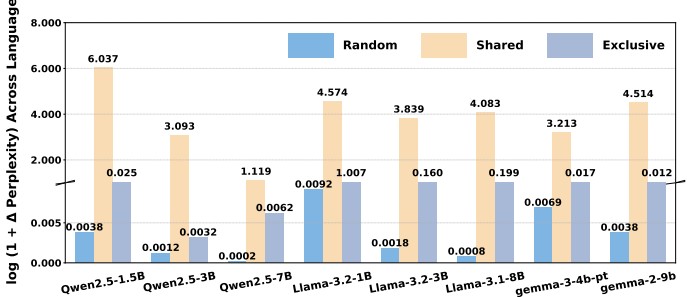

Figure 4: Perplexity changes caused by deactivating random neuron sets (Random), language-shared neurons (Shared) and language-exclusive neurons (Exclusive). Notice that Random deactivation barely affects models' perplexity, while Shared and Exclusive deactivation break the models' abilities.

**Deactivating shared and exclusive neurons both leads to model degradation.** Before contrasting language-shared neurons with language-exclusive neurons, we conduct a control experiment in

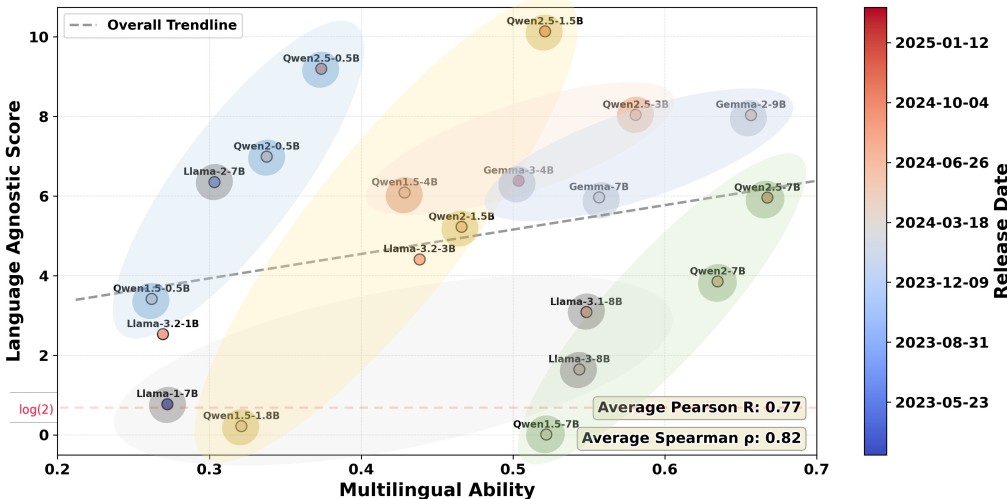

Figure 5: The relationship between multilingual ability and language-agnostic score (as defined in Equation 6) across various language models. Each point represents a model, colored by model size. The red dashed line (- - - -) indicates where shared neuron influence surpasses that of exclusive neurons, i.e., $\text{Imp} = 1$ and Language Agnostic Score $= \log(2)$. Shaded regions group models within the same series and of similar scale, while the dashed line (- - - -) indicates the overall trend: as LLMs evolve, successive generations with enhanced multilingual capabilities tend to achieve higher language-agnostic scores. This trend suggests that shared neurons increasingly support not only multilingual processing but also the emergence of more language-agnostic, abstract thought.

which we deactivate an equal number of randomly selected neurons, matching the quantity of both language-shared and language-exclusive neurons. As illustrated in Figure 4, this random deactivation results in minimal changes in perplexity across languages. In contrast, deactivating language-shared or language-exclusive neurons leads to significant performance degradation. These results confirm that the identified language-related neurons are indeed specialized for language processing, and that our neuron importance metrics are robust to random perturbations.

To further investigate whether language-shared neurons evolve into language-agnostic neurons, we analyze the evolution of the language-agnostic score, as defined in Equation 6, in relation to the models' multilingual capabilities, as shown in Figure 5.

**Shared neurons in early-stage models reflect superficial overlap without supporting higher-level cognition.** In earlier models such as the Qwen1.5 series and LLaMA-1-7B, deactivating shared neurons has a comparable effect to deactivating exclusive neurons, with language-agnostic scores around 1. This suggests that in early-stage models, shared neurons have the similar importance with exclusive neuron, and shared neurons largely reflect superficial overlaps between language-related neuron across languages, rather than representing a distinct, functionally meaningful shared space.

**Shared neurons in recent models become central and exhibit language-agnostic properties.** In contrast, recent models, such as those in the Qwen2.5 series, exhibit a dramatically different pattern. Deactivating shared neurons leads to a sharp and disproportionate increase in perplexity across all languages, often several orders of magnitude greater than the increase caused by removing language-exclusive neurons. In other words, shared neurons contribute far more critically to multilingual processing than exclusive neurons, despite both being part of the language-related neuron set. This disproportionate degradation reveals that shared neurons in recent models have evolved beyond serving merely as intersections of language-specific components; they now fulfill more fundamental, language-agnostic roles. If such shared neurons have indeed evolved into language-agnostic neurons, they may be operating within a conceptual space that abstracts away from surface-level linguistic variations. Such a space would allow the model to perform high-level reasoning, semantic alignment, and cross-lingual generalization—hallmarks of abstract thought in multilingual LLMs.

Table 1: Multilingual performance improvements on MGSM (primarily involving abstract thought) and MMMLU (requiring both abstract thought and domain knowledge) across five languages. Models were trained only on 100,000 general documents without reasoning-related data and evaluated using Llama-3.1-8B (high language-agnostic), Llama-3.2-3B (medium language-agnostic), and Llama-3.2-1B (low language-agnostic) under various targeted neuron tuning strategies.

| | Neuron | MGSM | | | | | | MMMLU | | | | | |
|---|---|---|---|---|---|---|---|---|---|---|---|---|---|
| | | Zh | Fr | De | Th | Sw | $\Delta_{Avg}$ | Zh | Fr | De | Th | Sw | $\Delta_{Avg}$ |
| Llama-3.1-8B | None | 52.4 | 51.6 | 54.4 | 46.8 | 38.8 | - | 53.8 | 58.4 | 56.9 | 48.8 | 40.9 | - |
| | Shared | $52.0^{-0.4}$ | $52.8^{+1.2}$ | $55.6^{+1.2}$ | $45.6^{-1.2}$ | $39.6^{+0.8}$ | 0.3 | $54.6^{+0.8}$ | $57.2^{-1.2}$ | $56.5^{-0.4}$ | $48.9^{+0.1}$ | $42.3^{+1.4}$ | 0.1 |
| | Exclusive | $56.8^{+4.4}$ | $57.2^{+5.6}$ | $57.2^{+2.8}$ | $50.4^{+3.6}$ | $42.4^{+3.6}$ | **4.0** | $55.6^{+1.8}$ | $59.2^{+0.8}$ | $59.1^{+2.2}$ | $49.9^{+1.1}$ | $43.7^{+2.8}$ | **1.7** |
| | Random | $50.4^{-2.0}$ | $51.2^{-0.4}$ | $54.4^{-0.0}$ | $47.2^{+0.4}$ | $37.6^{-1.2}$ | -0.6 | $52.4^{-1.4}$ | $58.3^{-0.1}$ | $57.2^{+0.3}$ | $47.1^{-1.7}$ | $41.3^{+0.4}$ | -0.5 |
| Llama-3.2-3B | None | 40.8 | 42.4 | 57.2 | 35.2 | 30.8 | - | 45.2 | 49.0 | 47.1 | 40.6 | 34.1 | - |
| | Shared | $42.8^{+2.0}$ | $45.6^{+3.2}$ | $66.4^{+9.2}$ | $40.4^{+5.2}$ | $39.6^{+8.8}$ | **5.7** | $44.9^{-0.3}$ | $49.8^{+0.8}$ | $47.3^{+0.2}$ | $41.0^{+0.4}$ | $34.8^{+0.7}$ | **0.4** |
| | Exclusive | $42.4^{+1.6}$ | $43.2^{+0.8}$ | $65.6^{+8.4}$ | $37.2^{+2.0}$ | $36.0^{+5.2}$ | 3.6 | $44.9^{-0.3}$ | $48.9^{-0.1}$ | $47.1^{+0.0}$ | $40.9^{+0.3}$ | $34.7^{+0.6}$ | 0.1 |
| | Random | $40.4^{-0.4}$ | $41.6^{-0.8}$ | $63.2^{+6.0}$ | $34.8^{-0.4}$ | $30.0^{-0.8}$ | 0.7 | $44.5^{-0.7}$ | $49.0^{+0.0}$ | $46.9^{-0.2}$ | $40.3^{-0.3}$ | $34.1^{+0.0}$ | -0.2 |
| Llama-3.2-1B | None | 26.4 | 26.0 | 29.2 | 20.0 | 22.8 | - | 29.0 | 27.8 | 28.8 | 28.8 | 26.6 | - |
| | Shared | $30.0^{+3.6}$ | $30.4^{+4.4}$ | $30.8^{+1.6}$ | $22.4^{+2.4}$ | $26.4^{+3.6}$ | 3.1 | $29.2^{+0.2}$ | $28.7^{+0.9}$ | $29.5^{+0.7}$ | $29.4^{+0.6}$ | $26.8^{+0.2}$ | **0.5** |
| | Exclusive | $27.6^{+1.2}$ | $30.0^{+4.0}$ | $34.4^{+5.2}$ | $23.2^{+3.2}$ | $30.4^{+7.6}$ | **4.2** | $29.0^{-0.0}$ | $28.0^{+0.2}$ | $29.3^{+0.5}$ | $28.2^{-0.6}$ | $26.8^{+0.2}$ | 0.1 |
| | Random | $26.8^{+0.4}$ | $26.4^{-0.4}$ | $29.6^{+0.4}$ | $21.2^{+1.2}$ | $26.4^{+3.6}$ | 1.0 | $28.8^{-0.2}$ | $28.3^{+0.5}$ | $29.1^{+0.3}$ | $28.6^{-0.2}$ | $26.8^{+0.2}$ | 0.1 |

# 4 Multilingual Enhancement via Neuron-Targeted Training

## 4.1 Language Agnostic Score Guided Multilingual Enhance

Inspired by above insights, we propose various targeted neuron training methods to enhance models' multilingual capability according to their language agnostic score.

**LLMs with low language agnostic score can train any language-related neurons.** These models exhibit limited multilingual capabilities, indicating that all language-related neurons require improvement. To enhance their performance across languages, we propose training all language-related neurons, whether they are shared across languages or specific to individual ones.

**LLMs with middle language agnostic score should train language-shared neurons.** These models demonstrate a degree of multilingual capability; however, the language-shared neurons have not yet evolved to become truly language-agnostic. Given that language-shared neurons are more prevalent than language-exclusive ones in these models, it is essential to further train and refine them to more effectively enhance the models' multilingual performance.

**LLMs with high language agnostic score should train language-exclusive neurons** The language-shared neurons in these models have evolved into language-agnostic neurons, responsible for abstract thought. They are already well-trained and offer limited room for further improvement. Therefore, to enhance multilingual performance, it is necessary to focus on training the language-exclusive neurons in these LLMs.

## 4.2 Experiment Setup

**Dataset** To further validate our hypothesis and explore how to utilize our findings to efficiently enhance multilingual capability in LLMs, we conduct continuous pretraining on specific neurons using multilingual corpora. Specifically, we construct a training set by sampling 100,000 examples per language from a mixture of three widely used multilingual datasets: Culturax (Nguyen et al., 2024), MADLAD (Kudugunta et al., 2023), and Wikipedia (Guo et al., 2020).

**Training Settings** We utilize Llama3.2-1B (Grattafiori et al., 2024), Llama3.2-3B, and Lamma-3.1-8B as representative LLMs with low, medium, and high language-agnostic scores, respectively. We conduct experiments under three training settings: language-shared neurons, language-exclusive neurons, and an equal number of randomly selected neurons. To evaluate multilingual capability, we employ the MMMLU and MGSM benchmarks.

**Experiment Results** Table 1 demonstrates that the multilingual capabilities of language models can be effectively enhanced through targeted neuron-specific tuning. For Llama-3.2-1B, which exhibits a relatively low language-agnostic score, tuning both shared and exclusive neurons significantly improves the model's cross-lingual reasoning performance, yielding average gains of 3.1 and 4.2 points on the MGSM benchmark, respectively. In the case of Llama-3.2-3B, which has a moderate language-agnostic score, tuning language-shared neurons results in the greatest performance improvement—an average gain of 5.7 points on MGSM. This is likely because these neurons are more numerous and less well-trained than exclusive ones. Finally, for Llama-3.2-8B, which already possesses a high language-agnostic score, the language-shared neurons appear to be sufficiently trained; thus, tuning exclusive neurons leads to further enhancement of multilingual performance, with an observed improvement of 4.0 points on MGSM. Compared to the improvement observed on MGSM, the performance gain on MMLU—which relies more heavily on knowledge extraction—is relatively smaller. This suggests that our training approach primarily enhances the model's thinking capabilities rather than its factual recall. Moreover, since we exclusively utilize general documents without incorporating reasoning-specific data, the substantial improvement further validates the effectiveness of our neuron-targeted training methodology. Additional experimental settings, results on different backbone LLMs, comparisons with baseline methods (e.g., LoRA), and cross-lingual evaluation analyses are provided in Appendix E and Appendix F.

## 5 Related Work

**Thinking Language of LLMs** Large language models (LLMs) (Touvron et al., 2023b; OpenAI, 2023; Zhang et al., 2024a; Chen et al., 2024; Liu et al., 2025c; Gemma Team et al., 2025) demonstrate strong multilingual reasoning and transfer abilities (Pires et al., 2019; Wu and Dredze, 2019; You et al., 2025; Cai et al., 2025; Nooralahzadeh et al., 2020), raising questions about whether these models operate in a language-agnostic or language-specific concept space (Nanda et al., 2023; Schut et al., 2025a; Zhao et al., 2024b), and which language would the model "think" in. One stream of work supports the hypothesis that LLMs "think" in a concept space centered on the predominant training language. Zhong et al. (2024) analyzed LLMs trained predominantly on English or Japanese (Fujii et al., 2024; LLM-jp et al., 2024) for their mainly activated languages; Fierro et al. (2025) showed language dependence in object retrieval; and Schut et al. (2025b), found representations align more closely with English even on foreign inputs. On the other hand, a language-agnostic view is supported by either probing studies (Pires et al., 2019; Stanczak et al., 2022), neuron-level manipulations (Dumas et al., 2024; Brinkmann et al., 2025; Ding et al., 2024) or both (Wu et al., 2025; Wendler et al., 2024). Our work falls in line with Dumas et al. (2024); Wendler et al. (2024); Wu et al. (2025), with more fine-grained neuron-level results and a novel activation-and-training-based analysis method.

**Multilingual Enhancement** Early-on, multilingual enhancement is mainly approached from pre-training in works such as XLM, XLM-R(Conneau et al., 2020; Lample and Conneau, 2019) and M-BERT (Devlin et al., 2018). More post-training work, ranging from continual pre-training (Zhang et al., 2021; Cui et al., 2024; Liu et al., 2025b; Husain et al., 2024; Kuulmets et al., 2024) to fine-tuning (Muennighoff et al., 2023; Chen et al., 2023; Ahuja et al., 2024; Lai et al., 2023; Indurthi et al., 2024; Lai and Nissim, 2024; Zhao et al., 2024c) have emerged to effectively improve models' multilingual abilities, though rather sensitive to training corpus and settings. A parallel body of work focuses on prompt-based methods, either leaning on language alignment (Zhang et al., 2024b; Etxaniz et al., 2023; Zhao et al., 2024a) or instruction-following and attention (Wang et al., 2025b; Zhao et al., 2025b,a). However, Liu et al. (2024) points out the suboptimality in translation-based prompting pipelines. Our neuron-specific tuning strategy answers the academic call (Liu et al., 2024, 2025a) for a more comprehensive approach to multilingual enhancement than translation-based prompting, and provides a more efficient and task-neutral alternative than the post-training based methods.

## 6 Conclusion & Discussion

In this work, we explore the emergence of abstract thought in large language models through the lens of neuron behavior. By identifying and categorizing language-related neurons as either shared or exclusive, we uncover a consistent trend across model development: shared neurons not only increase in proportion but also grow in functional importance, eventually forming a compact yet

critical set of language-agnostic neurons. These neurons underpin the model's ability to generalize across languages and support abstract reasoning that transcends linguistic boundaries. Motivated by this insight, we introduce neuron-specific training strategies that adapt to the developmental stage of an LLM, whether or not it exhibits language-agnostic behavior. Extensive experiments confirm that our targeted training approach effectively enhances multilingual performance across diverse models. We believe this neuron-centric perspective opens new avenues for understanding and improving the generalization capabilities of LLMs in multilingual and cross-lingual contexts.

Our study is limited by computational resources, which restricts evaluation on the larger LLMs and prevents full exploration of the potential of neuron-centric training at larger scales. We leave these directions for future work. Nonetheless, our findings shed light on the emergence of multilingual and abstract reasoning in LLMs, which may promote language equity but also raise risks like cross-lingual misinformation, calling for responsible deployment.

## Acknowledgements

This research is supported by the National Research Foundation, Singapore under its National Large Language Models Funding Initiative (AISG Award No: AISG-NMLP-2024-005) and National Large Language Models Funding Initiative (AISG Award No: AISG-NMLP-2024-002). Any opinions, findings and conclusions or recommendations expressed in this material are those of the authors and do not reflect the views of National Research Foundation, Singapore. This project was also partially supported by the Singapore Ministry of Education Academic Research Fund Tier 1 (Award Number: T1 251RES2514) and TPU Research Cloud.

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

# Appendix

# A  Parallel Neuron Detection Algorithm

Inspired by Zhao et al. (2024b); Wang et al. (2025a), the neuron detection method in Equation 2 can be done parallel. While Equation 2 considers the change in the final output embedding, the parallel methods described here efficiently calculate the change in the output of the *specific layer containing the neuron* when that neuron is deactivated. This layer-wise impact serves as a proxy or component for the overall impact.

In this context, let $X \in \mathbb{R}^{l \times d_{model}}$ be the input hidden states to a given layer, where $l$ is the sequence length and $d_{model}$ is the hidden dimension of the model. For a neuron $\mathcal{N}$ within this layer, its impact is measured as $\|f(X; \Theta) - f(X; \Theta_{\ominus \mathcal{N}})\|_2$, where $f(X; \Theta)$ is the layer's output with parameters $\Theta$, and $f(X; \Theta_{\ominus \mathcal{N}})$ is the output when neuron $\mathcal{N}$ (a specific row or column in $\Theta$) is deactivated (its parameters set to zero).

## A.1  Feed-Forward Network (FFN) Neurons

A standard FFN layer in modern transformer models can be expressed as:
$$\text{FFN}(X) = \left( \text{SiLU}(XW_{gate}) \odot (XW_{up}) \right) W_{down} \tag{7}$$
where $X \in \mathbb{R}^{l \times d_{model}}$ is the input to the FFN layer, $W_{gate}, W_{up} \in \mathbb{R}^{d_{model} \times d_{inter}}$, and $W_{down} \in \mathbb{R}^{d_{inter} \times d_{model}}$. Here, $d_{inter}$ is the intermediate dimension of the FFN. The symbol $\odot$ denotes element-wise multiplication. Let $H_{act} = \text{SiLU}(XW_{gate}) \odot (XW_{up})$ be the intermediate activation matrix, $H_{act} \in \mathbb{R}^{l \times d_{inter}}$. Thus, the FFN output is $Y_{FFN} = H_{act} W_{down} \in \mathbb{R}^{l \times d_{model}}$.

We consider a neuron $\mathcal{N}_{inter,k}$ to be associated with the $k$-th dimension of the intermediate representation $H_{act}$. Deactivating such a neuron means that the $k$-th column of $H_{act}$, denoted $H_{act}[:, k]$, is effectively zeroed out before the multiplication with $W_{down}$. This deactivation corresponds to zeroing out the parameters that produce this $k$-th intermediate feature, e.g., the $k$-th column of $W_{up}$ (i.e., neuron $\mathcal{N}$ is $W_{up}[:, k]$) and $W_{gate}$, or by zeroing out parameters that read from it, e.g., the $k$-th row of $W_{down}$ (i.e., neuron $\mathcal{N}$ is $W_{down}[k, :]$).

Let $Y_{FFN, \ominus \mathcal{N}_{inter,k}}$ be the output when the $k$-th intermediate neuron is deactivated. The change in the layer's output is:
$$\Delta Y_{FFN,k} = Y_{FFN} - Y_{FFN, \ominus \mathcal{N}_{inter,k}}$$
If $H'_{act}$ is $H_{act}$ with its $k$-th column zeroed, then $Y_{FFN, \ominus \mathcal{N}_{inter,k}} = H'_{act} W_{down}$. So,
$$\Delta Y_{FFN,k} = (H_{act} - H'_{act}) W_{down}$$
The matrix $(H_{act} - H'_{act})$ is zero everywhere except for its $k$-th column, which consists of the elements $H_{act}[:, k]$. Let this difference matrix be $\delta H_k$. Then $\Delta Y_{FFN,k} = \delta H_k W_{down}$. This resulting $l \times d_{model}$ matrix is formed by the outer product of the $k$-th column of $H_{act}$ and the $k$-th row of $W_{down}$:
$$\Delta Y_{FFN,k} = H_{act}[:, k] (W_{down})_{k,:}$$
The impact of the $k$-th intermediate FFN neuron is then the L2 norm of this change:
$$\|\Delta Y_{FFN,k}\|_2 = \|H_{act}[:, k] (W_{down})_{k,:}\|_2 \tag{8}$$
This computation can be performed in parallel for all $k \in \{1, \ldots, d_{inter}\}$ to obtain the impact of all intermediate neurons in the FFN layer.

## A.2  Self-Attention Network Neurons

The output of a self-attention layer (for simplicity, we describe a single attention head; multi-head attention involves similar computations per head) can be given by:
$$Y_{Attn} = \text{Softmax}\left( \frac{(XW_Q)(XW_K)^T}{\sqrt{d_k}} \right) (XW_V) \tag{9}$$

Let $Q = XW_Q \in \mathbb{R}^{l \times d_{attn}}$, $K = XW_K \in \mathbb{R}^{l \times d_{attn}}$, and $V = XW_V \in \mathbb{R}^{l \times d_{attn}}$, where $d_{attn}$ is the dimension of queries, keys, and values for the attention mechanism. $d_k$ is the scaling factor, typically the dimension of the key/query vectors (e.g., $d_k = d_{attn}$). Let $A = \text{Softmax}\left( \frac{QK^T}{\sqrt{d_k}} \right) \in \mathbb{R}^{l \times l}$. The layer output is $Y_{Attn} = AV \in \mathbb{R}^{l \times d_{attn}}$. (An additional output projection $W_O$ might follow this, which would be multiplied subsequently).

### A.2.1 Neurons in $W_V$

Consider a neuron $\mathcal{N}_{V,k}$ defined as the $k$-th column of $W_V$, i.e., $W_V[:,k]$. Deactivating this neuron sets $W_V[:,k]$ to zero, which in turn makes the $k$-th column of $V = XW_V$, denoted $V[:,k]$, zero. Let $V'$ be the matrix $V$ with its $k$-th column zeroed. The change in the layer's output is:

$$\Delta Y^{(V)}_{Attn,k} = AV - AV' = A(V - V')$$

The matrix $(V - V')$ is zero everywhere except for its $k$-th column, which is $V[:,k]$. Let this difference matrix be $\delta V_k$. Then $\Delta Y^{(V)}_{Attn,k} = A(\delta V_k)$. This $l \times d_{attn}$ matrix has $AV[:,k]$ (the matrix $A$ multiplied by the vector $V[:,k]$) as its $k$-th column, and zeros in other columns. The impact of neuron $\mathcal{N}_{V,k}$ is:

$$\left\| \Delta Y^{(V)}_{Attn,k} \right\|_2 = \| AV[:,k] \|_2 \tag{10}$$

where the norm is effectively taken over the $l \times 1$ vector $AV[:,k]$ that forms the $k$-th column of the change matrix. This can be calculated in parallel for all $k \in \{1, \ldots, d_{attn}\}$.

### A.2.2 Neurons in $W_Q$

Consider a neuron $\mathcal{N}_{Q,k}$ defined as the $k$-th column of $W_Q$, i.e., $W_Q[:,k]$. Deactivating this neuron sets $W_Q[:,k]$ to zero. This makes the $k$-th column of $Q = XW_Q$, denoted $Q[:,k]$, zero. Let $Q'$ be the matrix $Q$ with its $k$-th column zeroed. The original unnormalized attention scores are $S_{raw} = \frac{QK^T}{\sqrt{d_k}}$. The new unnormalized attention scores with $\mathcal{N}_{Q,k}$ deactivated are $S'_{raw} = \frac{Q'K^T}{\sqrt{d_k}}$. The change in the unnormalized scores due to deactivating $\mathcal{N}_{Q,k}$ is $\Delta S_{raw,k} = S_{raw} - S'_{raw} = \frac{(Q-Q')K^T}{\sqrt{d_k}}$. The matrix $(Q - Q')$ is zero everywhere except for its $k$-th column, which is $Q[:,k]$. Thus,

$$\Delta S_{raw,k} = \frac{(Q[:,k])(K[:,k])^T}{\sqrt{d_k}}$$

This $l \times l$ matrix represents the change in raw attention scores attributable to the interaction involving the $k$-th column of $Q$ and the $k$-th column of $K$.

Let $A_{orig} = \text{softmax}(S_{raw})$ be the original attention probability matrix. Let $A_{\ominus \mathcal{N}_{Q,k}} = \text{softmax}(S_{raw} - \Delta S_{raw,k})$ be the attention probability matrix when neuron $\mathcal{N}_{Q,k}$ is deactivated. The change in the layer's output is:

$$\Delta Y^{(Q)}_{Attn,k} = A_{orig}V - A_{\ominus \mathcal{N}_{Q,k}}V = (A_{orig} - A_{\ominus \mathcal{N}_{Q,k}})V$$

The impact of neuron $\mathcal{N}_{Q,k}$ is:

$$\left\| \Delta Y^{(Q)}_{Attn,k} \right\|_2 = \left\| (A_{orig} - A_{\ominus \mathcal{N}_{Q,k}})V \right\|_2 \tag{11}$$

To calculate this efficiently for all $k \in \{1, \ldots, d_{attn}\}$ (corresponding to each column neuron in $W_Q$):

1. Compute the original $S_{raw} = \frac{QK^T}{\sqrt{d_k}}$ and $A_{orig} = \text{softmax}(S_{raw})$.

2. For each $k$, compute the specific change term $\Delta S_{raw,k} = \frac{Q[:,k](K[:,k])^T}{\sqrt{d_k}}$. This step can be parallelized by constructing a tensor $\Delta S_{raw} \in \mathbb{R}^{d_{attn} \times l \times l}$ where the slice $\Delta S_{raw}[k,:,:] = \Delta S_{raw,k}$.

3. For each $k$, compute the adjusted scores $S_{adjusted,k} = S_{raw} - \Delta S_{raw}[k,:,:]$.

4. For each $k$, compute $A_{\ominus \mathcal{N}_{Q,k}} = \text{softmax}(S_{adjusted,k})$.

5. For each $k$, calculate the impact norm $\| (A_{orig} - A_{\ominus \mathcal{N}_{Q,k}})V \|_2$.

### A.2.3 Neurons in $W_K$

The impact of deactivating a neuron $\mathcal{N}_{K,k}$ (the $k$-th column of $W_K$) is calculated symmetrically to that of $\mathcal{N}_{Q,k}$. The same change term $\Delta S_{raw,k} = \frac{Q[:,k](K[:,k])^T}{\sqrt{d_k}}$ is used, reflecting the idea that this term captures the interaction component associated with the $k$-th features of both $Q$ and $K$. The procedure then follows steps 3-5 as outlined for $W_Q$ neurons, using this $\Delta S_{raw,k}$ to find the adjusted attention matrix and the resulting impact.

# B  Neuron Detection Threshold

An important implementation detail in identifying language-related neurons lies in the selection of the activation threshold $\sigma$. Rather than adopting a fixed global scalar, we employ a dynamic thresholding mechanism that adapts to each query. Specifically, as shown in our released implementation, for every query in a given language, we rank neurons based on their computed importance scores and select the top $1\%$ as activated neurons. Subsequently, for each language $\ell$, its language-specific neuron set $\mathcal{N}_{\text{lang}}^{\ell}$ is defined as the intersection of these top-ranked neurons across all queries belonging to that language.

This dynamic top-$1\%$ strategy ensures consistent sensitivity across languages with different overall activation magnitudes, allowing the model to capture meaningful variations without being biased by language-specific activation scales. The choice of $1\%$ is empirically determined through a set of calibration experiments designed to balance selectivity and stability.

To validate the appropriateness of this threshold, we conduct a sanity check using random baselines. For each language, we compare the model degradation caused by deactivating the selected language-specific neurons with that caused by deactivating an equal number of randomly chosen neurons. In all cases, we observe that removing the identified neurons results in a drastic performance drop—often exceeding a $100\times$ increase in perplexity—while removing random neurons yields negligible effects. This substantial performance disparity confirms that the selected neurons are functionally meaningful and that the threshold effectively distinguishes critical neurons from background noise.

If, conversely, random neuron deactivation were to cause a comparable decline in performance, it would suggest that the threshold is too lenient, allowing excessive neurons to be classified as important. In such cases, the percentile threshold would be systematically reduced until the random baseline no longer impacts model behavior. This adaptive validation process ensures that the threshold $\sigma$ remains both rigorous and empirically grounded across all examined languages and model families.

# C  Neuron Analysis

To further understand the internal organization of multilingual representations within large language models, we conduct a comprehensive neuron-level analysis. This section explores how language-shared, language-exclusive, and strictly unique neurons are distributed across different architectural components and model layers, offering insight into how multilingual models balance generalization and specialization.

## C.1  Component-Level Distribution

We first analyze how neurons are distributed across major architectural components. Neurons are categorized into three groups: query-key (QK), value-output (VO), and feed-forward network (FFN). As shown in Table 2, shared neurons are primarily concentrated in the QK components, aligning with the general attention mechanism responsible for capturing cross-lingual relational patterns. In contrast, exclusive neurons are more prevalent in VO and FFN layers, indicating their more important role in language-specific transformations and output generation. This decomposition suggests that shared and exclusive neurons perform complementary roles in multilingual processing.

We find that the component-level trend remains consistent across model families: shared neurons concentrate within the attention's QK submodules, supporting cross-lingual abstraction, while exclusive neurons appear more prominently in VO and FFN blocks, handling language-specific representations and refinements.

## C.2  Layer-wise Distribution

We further examine how these neurons are distributed across model layers. Figures 6 and 7 illustrate the proportion of shared and exclusive neurons across all layers for Gemma2-9B and LLaMA3.1-8B, respectively. To quantify the trend, we group layers into early, middle, and late stages and report the average proportions in Table 3.

The layer-wise distribution reveals distinct allocation patterns for shared and exclusive neurons. Exclusive neurons are more concentrated in the early and late layers, suggesting that language-

Table 2: Component-level neuron distribution across models. Percentages represent the proportion of shared or exclusive neurons (averaged across languages) within each component.

| Model | Component | Shared (%) | Exclusive (%) |
|---|---|---|---|
| LLaMA3.1-8B | QK | 92.50 | 59.48 |
| | VO | 2.90 | 23.80 |
| | FFN | 4.59 | 16.71 |
| Qwen2.5-3B | QK | 76.40 | 53.06 |
| | VO | 16.72 | 30.36 |
| | FFN | 6.88 | 16.58 |
| Gemma2-9B | QK | 65.32 | 35.40 |
| | VO | 22.02 | 40.46 |
| | FFN | 12.66 | 24.16 |

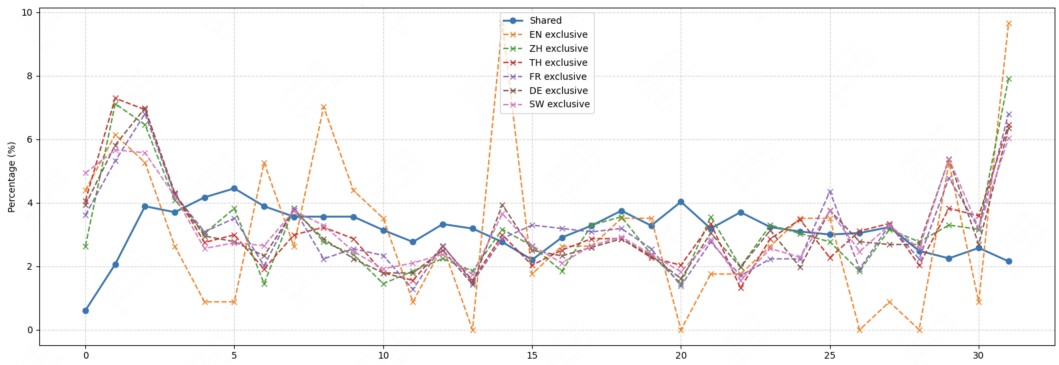

Figure 6: Layer-wise distribution analysis of Gemma2-9B.

specific processing primarily occurs near the input and output boundaries, where lexical and syntactic variations are handled. In contrast, shared neurons maintain a stable proportion across all layers, reflecting their role in capturing transferable, cross-lingual representations throughout the network.

Overall, these analyses demonstrate that while both neuron types are crucial for multilingual processing, shared neurons form a stable representational backbone that supports language-agnostic reasoning, whereas exclusive neurons enable fine-grained, language-specific adjustments near the model periphery. These findings are consistent across LLaMA, Qwen, and Gemma series, reinforcing the robustness of this observation.

## C.3 Unique Language Neurons

To further refine our understanding of neuron selectivity, we investigate strictly unique neurons—those that respond exclusively to one language. While our definition of language-exclusive neurons allows for activation across a subset of languages, this stricter criterion provides additional insight into the specialization of multilingual models.

Table 4 reports the proportion of strictly unique neurons per language. These neurons constitute only a small percentage of the total population, suggesting that the model predominantly relies on shared neurons for multilingual understanding. Interestingly, higher values for Swahili and Thai—both lower-resource languages—indicate a stronger reliance on language-specific neurons, likely to compensate for limited training data.

We also analyze their layer-wise distribution by grouping the model layers into three stages: early (0–7), middle (8–23), and late (24–31). The averaged proportions of unique, shared, and exclusive neurons in each group are shown in Table 5. The early and late stages exhibit higher proportions of unique and exclusive neurons, implying that language-specific encoding and decoding occur near the input and output boundaries. The middle layers, dominated by shared neurons, correspond to a cross-lingual abstraction stage responsible for language-independent reasoning.

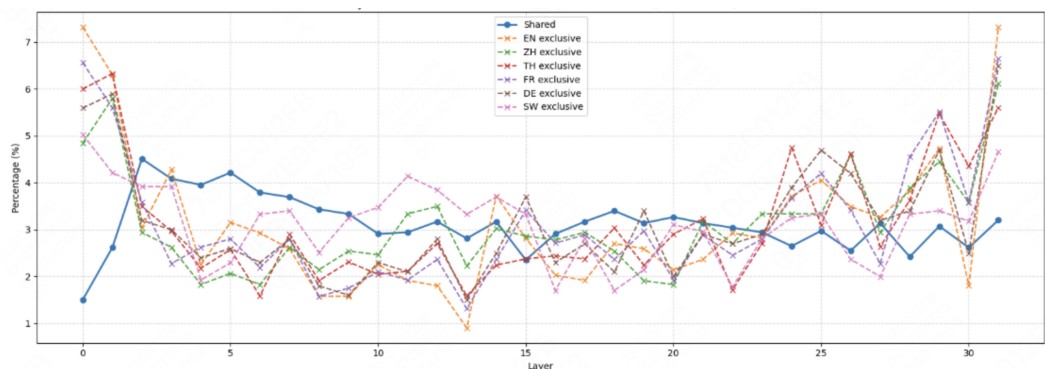

Figure 7: Layer-wise distribution analysis of LLaMA3.1-8B.

Table 3: Layer-wise neuron distribution across representative models. Percentages indicate average proportions of shared and exclusive neurons within each layer range.

| Model | Layer Range | Shared (%) | Exclusive (%) |
|---|---|---|---|
| LLaMA3.1-8B | 0–7 | 3.07 | 4.40 |
| | 8–23 | 3.11 | 2.89 |
| | 24–31 | 2.83 | 3.83 |
| Qwen2.5-3B | 0–5 | 3.69 | 4.45 |
| | 6–29 | 2.70 | 2.43 |
| | 30–35 | 2.20 | 2.48 |
| Gemma2-9B | 0–5 | 2.41 | 3.13 |
| | 6–34 | 2.39 | 1.92 |
| | 35–41 | 2.29 | 3.65 |

Finally, we examine the component-level distribution of strictly unique neurons (Table 6). Shared neurons cluster within QK, reinforcing their role in cross-lingual alignment. Exclusive neurons, as well as strictly unique ones, are more concentrated in VO and FFN, highlighting their specialization in language-specific semantic transformation and output projection.

These findings reinforce our broader conclusion: shared neurons underpin cross-lingual generalization, capturing transferable semantics; exclusive neurons encode language-specific nuances, crucial for accurate understanding and generation; and strictly unique neurons—being the most selective subset—reflect the model's fine-grained adaptation to individual languages, typically concentrated at the model's periphery where input encoding and output generation occur.

# D  Neuron Detection Corpus

This section provides additional details regarding the corpus used for neuron detection, as mentioned in the main text. Our methodology relies on the OSCAR corpus for both identifying language-related neurons through activation patterns and quantifying their functional contribution via perplexity changes upon deactivation.

## D.1  OSCAR Corpus

The OSCAR (Open Super-large Crawled Aggregated coRpus) corpus (Abadji et al., 2022) is a massive multilingual collection of texts obtained by language classification and filtering of the Common Crawl dataset. Common Crawl is a publicly available web crawl spanning petabytes of data. OSCAR further processes this raw data to produce monolingual corpora across a wide range of languages, making it a valuable resource for training large language models and conducting cross-lingual research.

Key characteristics of the OSCAR corpus include:

• **Large Scale:** It contains hundreds of gigabytes to terabytes of text data for many languages.

Table 4: Proportion of strictly unique neurons (responding to only one language).

| Language | Strictly Unique Neurons (%) |
|---|---|
| English (en) | 0.02 |
| Chinese (zh) | 0.03 |
| Thai (th) | 0.06 |
| French (fr) | 0.03 |
| German (de) | 0.02 |
| Swahili (sw) | 0.07 |

Table 5: Layer-wise distribution of unique, shared, and exclusive neurons in LLaMA3.1-8B.

| Layer Group | Unique (%) | Shared (%) | Exclusive (%) |
|---|---|---|---|
| Early (0–7) | 3.69 | 3.07 | 4.40 |
| Middle (8–23) | 2.28 | 3.11 | 2.89 |
| Late (24–31) | 4.62 | 2.83 | 3.83 |

- **Multilingual Coverage:** It supports a vast number of languages, facilitating studies that require diverse linguistic data.

- **Data Cleaning:** Efforts are made to clean and filter the crawled data, though the quality can vary depending on the language and the nature of web content.

- **Accessibility:** OSCAR is publicly available, promoting reproducibility and broader research in NLP.

For our study, we sample 1000 sentences for each target language from its respective monolingual section within the OSCAR corpus. This sampled data serves as the basis for analyzing neuron activations and evaluating perplexity changes. The diversity and scale of OSCAR help in capturing a wide array of linguistic phenomena necessary for robustly identifying language-specific neural correlates.

### D.2 Illustration of Sample Sentences

To provide a concrete illustration of the data used, Table 7 presents conceptual example sentences from the OSCAR corpus for the five languages central to our analysis: English (en), Chinese (zh), Swahili (sw), German (de), and French (fr).

The sentences sampled for each language are then further used to observe which neurons are consistently activated during processing. A similar set of sentences is then used to measure the perplexity of the model when specific neurons or sets of neurons are deactivated, thereby quantifying their functional importance to that language.

## E    Multilingual Enhancement

This section provides additional details on our multilingual enhancement experiments, including (1) the construction of random neuron baselines, (2) results on additional backbones, and (3) comparisons with LoRA, a famous parameter-efficient fine-tuning method. Together, these analyses validate the robustness and efficiency of our neuron-level enhancement strategy.

### E.1    Illustration of Random Neurons

Regarding the random neurons presented in Figure 4, we carefully designed the sampling process to ensure a fair comparison. Specifically, we sampled two sets of random neurons—each matching the total number of shared and exclusive neurons, respectively. These random neurons were uniformly sampled across all layers and components of the model, under the assumption of a homogeneous distribution.

Table 6: Component-level comparison among unique, shared, and exclusive neurons in LLaMA3.1-8B.

| Component | Unique (%) | Shared (%) | Exclusive (%) |
|-----------|-----------|-----------|--------------|
| QK | 18.40 | 92.50 | 59.48 |
| VO | 50.96 | 2.90 | 23.80 |
| FFN | 30.64 | 4.59 | 16.71 |

Table 7: Illustrative sample sentences from the OSCAR corpus for the selected languages. These are conceptual examples, as actual sentences are randomly sampled.

| Language | Conceptual Example |
|----------|-------------------|
| English (en) | The quick brown fox jumps over the lazy dog. |
| Chinese (zh) | 敏捷的棕色狐狸跳过了懒惰的狗。 |
| Swahili (sw) | Mbweha mwepesi wa kahawia anaruka juu ya mbwa mvivu. |
| German (de) | Der schnelle braune Fuchs springt über den faulen Hund. |
| French (fr) | Le renard brun rapide saute par-dessus le chien paresseux. |

We then selected the random set that exhibited a stronger influence on model performance, and used it consistently in both Figure 4 and Table 1. This ensures comparability across evaluations. It is indeed expected that the distribution and activation patterns of random neurons differ from those of the identified language-specific neurons, as the latter capture semantically grounded linguistic features rather than arbitrary activation patterns.

## E.2 Results on Additional Backbones

Figure 1 presents a simplified cross-model analysis within the same generation but across different model sizes. We observe that larger models generally exhibit a lower proportion and reduced importance of shared neurons. This aligns with prior findings showing that as models scale up, parameter specialization increases, leading to fewer neurons being shared across languages.

Although our main focus lies in analyzing the evolution of abstract thought over model development rather than size scaling, we include this discussion for completeness. To further verify the generality of our findings, we conducted additional analyses on the Qwen-2.5 family, particularly the 1.5B variant. The results are summarized below:

Table 8: Performance comparison of shared, exclusive, and random neuron sets on Qwen-2.5-1.5B. Metrics represent accuracy (%) on MGSM and MMMLU datasets.

| Model | MGSM | | | | | | MMMLU | | | | | |
|-------|------|-----|-----|-----|-----|---|-------|-----|-----|-----|-----|---|
| | zh | de | fr | th | sw | $\Delta$ | zh | de | fr | th | sw | $\Delta$ |
| Qwen2.5-1.5B | 63.60 | 57.20 | 61.20 | 50.80 | 28.00 | – | 53.95 | 48.47 | 50.96 | 44.00 | 30.49 | – |
| + Exclusive | 65.20 | 57.60 | 61.60 | 52.00 | 29.60 | +1.04 | 53.81 | 48.65 | 51.29 | 44.49 | 31.57 | +0.39 |
| + Shared | 63.20 | 56.80 | 62.00 | 49.20 | 31.20 | +0.32 | 53.72 | 48.38 | 51.30 | 44.42 | 31.48 | +0.29 |
| + Random | 62.80 | 54.00 | 60.40 | 46.80 | 27.60 | -2.88 | 53.71 | 48.62 | 51.09 | 44.17 | 31.36 | -0.17 |

These results indicate that both shared and exclusive neuron adjustments consistently improve multilingual reasoning, whereas random neuron updates negatively affect performance.

We further tested our findings on the Gemma2-9B model, which represents a different model family and a high language-agnostic score at the global level. The results are presented in Table 9.

We observe consistent trends across backbones: fine-tuning exclusive neurons enhances reasoning in target languages, while shared neurons contribute to general stability. In contrast, random or unstructured modifications fail to improve multilingual alignment.

Furthermore, our GSM8K experiments confirm that fine-tuning a model on a specific language improves reasoning in that language but may degrade performance in others—supporting our hypothesis that language-specific adaptation often comes at the cost of reduced cross-lingual transferability.

Table 9: Performance of neuron subsets on Gemma2-9B across languages.

| Subset | Zh | Fr | De | Th | Sw | Avg Δ |
|--------|------|------|------|------|------|-------|
| None | 58.4 | 58.0 | 58.8 | 57.2 | 51.2 | – |
| Shared | 56.8 | 57.6 | 58.8 | 54.8 | 48.4 | -1.4 |
| Exclusive | 61.6 | 60.8 | 62.4 | 58.4 | 55.6 | +3.0 |
| Random | 56.0 | 57.6 | 57.2 | 56.0 | 50.8 | -1.2 |

## E.3 Comparison with LoRA

Although our method focuses on neuron-level tuning rather than introducing new fine-tuning layers, we further conduct a comparison with LoRA to verify the efficiency and effectiveness of our approach. Unlike LoRA and other parameter-efficient fine-tuning (PEFT) methods, which insert additional adapter layers and train extra parameters, our method adjusts only a small subset of existing shared neurons. This enables multilingual enhancement without any architectural modification or parameter growth, highlighting a distinct trade-off between targeted internal adaptation and parameter-efficient extension.

For fair comparison, we implement LoRA-based fine-tuning on LLaMA3.2-3B under a similar parameter budget (rank = 48). As shown in Table 10, LoRA improves performance in certain MGSM cases (e.g., German, Swahili) but shows limited generalization and even degradation on MMMLU. Moreover, LoRA requires longer training time (2.2 hours on 2×H200 GPUs) compared to our neuron-level method (1.5 hours), demonstrating that our approach achieves competitive multilingual gains with higher efficiency and simpler implementation.

Table 10: Comparison between our neuron-level fine-tuning and LoRA on LLaMA3.2-3B.

| Model | MGSM | | | | | | MMMLU | | | | | |
|-------|------|------|------|------|------|------|------|------|------|------|------|------|
| | zh | de | fr | th | sw | Δ | zh | de | fr | th | sw | Δ |
| LLaMA3.2-3B | 40.80 | 57.20 | 42.40 | 35.20 | 30.80 | – | 45.20 | 47.10 | 49.00 | 40.60 | 34.10 | – |
| + LoRA | 38.80 | 68.80 | 44.00 | 31.20 | 37.20 | +2.72 | 44.40 | 45.79 | 47.74 | 39.58 | 32.50 | -1.12 |

Overall, LoRA demonstrates partial improvements but lacks consistency across benchmarks and languages. In contrast, our neuron-level approach achieves stable multilingual enhancement with lower computational overhead, highlighting its simplicity and interpretability as a complementary direction to PEFT methods.

## F Cross-lingual Evaluation

To further explore the effect of language-specific fine-tuning on multilingual generalization, we conduct cross-lingual evaluation experiments. While our primary focus is on understanding how fine-tuning with a single language corpus can enhance performance in that language, it is equally important to assess how such adaptation influences the model's capabilities across other languages.

We fine-tune LLaMA-3.2-3B using corpora from individual languages and then evaluate its performance on all target languages. Two metrics are examined: (1) accuracy improvement on the multilingual GSM8K (MGSM) benchmark, reflecting reasoning capability; and (2) change in language perplexity (PPL), reflecting language understanding and fluency.

Table 11 reports the results of fine-tuning on one language and testing across all others. "Target Language Acc" denotes the improvement on the language used for fine-tuning, while "Other Languages Acc (Avg)" shows the average accuracy change over the remaining four languages.

We also measure the change in perplexity (PPL) before and after fine-tuning as an intuitive indicator of the model's linguistic understanding. A decrease in PPL indicates improved fluency and comprehension in that language. The results are summarized in Table 12.

These results demonstrate a consistent trade-off: fine-tuning on a single language improves both reasoning ability and linguistic understanding in that language, but often at the expense of reduced performance on others. This observation suggests that language-specific adaptation repurposes part

Table 11: Cross-lingual evaluation on MGSM for LLaMA-3.2-3B. Fine-tuning on a single language improves reasoning in that language but moderately reduces performance in others.

| Language Trained On | Target Language Acc | Other Languages Acc (Avg) |
|---|---|---|
| Chinese (Zh) | +2.0 | -1.6 |
| French (Fr) | +3.2 | -2.4 |
| German (De) | +9.2 | -3.2 |
| Thai (Th) | +5.2 | -2.8 |
| Swahili (Sw) | +8.8 | -2.0 |

Table 12: Change in language perplexity (PPL) before and after fine-tuning on LLaMA-3.2-3B. Negative values indicate reduced perplexity (better language modeling).

| Language Trained On | Target Language PPL | Other Languages PPL (Avg) |
|---|---|---|
| Chinese (Zh) | -3.15 | +0.89 |
| French (Fr) | -2.03 | +0.77 |
| German (De) | -2.76 | +0.64 |
| Thai (Th) | -0.41 | +0.75 |
| Swahili (Sw) | -12.40 | +1.84 |

of the shared neuron subspace to better align with the target language, consequently weakening cross-lingual generalization.

In summary, language-specific fine-tuning enhances targeted capabilities while moderately compromising multilingual balance, implying that shared neurons serve as a critical mechanism for maintaining cross-lingual consistency.

# G Broder Impacts

This work contributes to a deeper understanding of how LLMs develop multilingual and abstract reasoning capabilities, which may help improve language equity in AI systems. By enhancing performance across diverse languages, our methods could benefit underrepresented linguistic communities. However, stronger multilingual models also carry risks, such as enabling more sophisticated misinformation in multiple languages. We encourage responsible use and further research into safeguards for multilingual LLM deployment.

# H Limitations

While our study provides compelling evidence for the emergence of abstract, language-agnostic thought in LLMs and demonstrates the effectiveness of neuron-centric training strategies, several limitations remain: First, due to resource constraints, our analysis—though conducted across diverse model families and scales—has not been extended to the larger LLMs. Whether the observed patterns of neuron sharing and functional importance generalize to such scales remains an open question. Second, although our proposed training strategy yields consistent gains, the scope of our experiments remains limited by computational cost. We have not fully explored the upper bound of performance improvements that could be achieved with larger-scale or longer-term neuron-centric training. We leave these limitations as future work.

