# OpenReview forum: "The Emergence of Abstract Thought in Large Language Models Beyond Any Language"
_NeurIPS.cc/2025/Conference — NeurIPS 2025 poster_

### Official Review · Reviewer_ysL3 · 2025-06-30

**Clarity:** 4
**Significance:** 4
**Originality:** 4
**Rating:** 5
**Confidence:** 3

**Summary:**

This work investigates language-shared and language-specific neurons in multilingual LLMs. The authors identify neurons whose removal has a significant impact on the resulting embeddings. For some of these neurons, the impact is apparent for all languages, while for other neurons, the effect is not specific to a (subset of) language(s). The authors define a "language agnostic score" score that quantifies the impact (in terms of changes in perplexity) of language-shared neurons.

The analyses, performed with several families of models (qwen , llama, gemma), reveal that the proportions of language-selective and language-shared neurons are small (around 0.3%).  Morevover, the relative proportions of shared vs selective neurons increases with the model generation and performance on a multilingual benchmark. The language agnostic score, that is, the importance of the shared-neurons, also increases with multilingual ability. In recent models, deactivating shared neurons increases perplexity more than deactivating language-exlusive neurons (Fig.4)

Finally, the authors further trained language--shared and language-exclusive neurons in models of the Llama family, and assess them on the Multi-task Language Understanding (MMMLU)  and Multilingual Grade School Math (MGSM) benchmarks. They report that the model with the highest agnostic score benefits more from training language-specific neurons, while the model with the  lowest agnostic scores benefits from training both share and exclusive neurons.  These leads them to claim that the language-shared neurons support abstract thinking.

**Questions:**

- For a given release date,  the proportion of shared neurons and their importance decrease with model size  in the qwen 2.5 family while the multilingual ability increases with model size.  Any idea why? Would the analysis of section 4 performed on the llama-3.2 family apply to qwen-2.5?

-  the number of language sensitive neurons depends on the threshold \sigma  (eq.1) but I did not see in the paper how this theshold was set/determined (?)

- The proposed definition of "language-exclusive neurons" (eq.3) includes neurons that may respond to several languages, but not all of them (which belong to N_shared). Another possibility would have been to "language-selective neurons as neuron that respond to only to one language". It could be nice to have the distribution of count of neurons responding  to 1, 2, 3, ...n language.

-   The conclusion/discussion is somewhat short.  As the authors claim that their study  shows the emergence of "abstract thought", and not solely of language agnostic neurons, it would be nice to reiterate the arguments spread in the paper and discuss their potential limitations.

**Ethical Concerns:**

["NO or VERY MINOR ethics concerns only"]

**Final Justification:**

The first version of the paper was quite good paper, and the clarifications, especially the crucial one about the determination of the sigma threshold, are convincing.  I therefore raised the scores to 4.

**Limitations:**

yes

**Quality:**

4

**Strengths And Weaknesses:**

The paper reads quite well. The authors' convincingly demonstrate that the proportion and importance of *language-agnostic* neurons increase with the  LLMs  generation (although  not systematically with model size or multilingual ability within a generation).

The approach of focusing the training on subsets of neurons (language-speicfic or language agnostic one) is particularly original.

A more thorough investigation of "abstract thought" with more benchmarks dissociating between linguistic and reasoning abilities, could have strengthened the conclusion that this is evidence for the emergence of "abstract thought".

---

> ### Author Rebuttal · Authors · 2025-07-31
>
> We thank the reviewer for the thorough and valuable feedback. To address your concerns, we present the point-to-point responses as follows.
>
> > **Comment 1:  More benchmark.** — “ A more thorough investigation of "abstract thought" with more benchmarks …”
> >
>
> Thank you for the insightful suggestion. We fully agree that distinguishing between linguistic and reasoning abilities is essential for substantiating our hypothesis regarding the emergence of abstract thought.
>
> To address this, we evaluated the model from both perspectives:
>
> 1. **Linguistic Benchmark:** We used **perplexity** as a direct metric to assess the model’s language understanding.
> 2. **Reasoning Benchmark:** We measured performance on **GSM8K**, a reasoning-intensive task.
>
> Specifically, we fine-tuned LLaMA-3.2-3B on single-language corpora and tested the resulting models across all five target languages. The results are summarized below:
>
> **GSM8K Accuracy (Reasoning Ability):**
>
> | Language Trained On | Target Language Acc | Other Languages Acc (Avg) |
> | --- | --- | --- |
> | Zh | +2.0 | -1.6 |
> | Fr | +3.2 | -2.4 |
> | De | +9.2 | -3.2 |
> | Th | +5.2 | -2.8 |
> | Sw | +8.8 | -2.0 |
>
> **Language Perplexity (Linguistic Understanding):**
>
> | Language Trained On | Target Language PPL  | Other Languages PPL (Avg) |
> | --- | --- | --- |
> | Zh | -3.15 | +0.89 |
> | Fr | -2.03 | +0.77 |
> | De | -2.76 | +0.64 |
> | Th | -0.41 | +0.75 |
> | Sw | -12.4 | +1.84 |
>
> These results clearly dissociate the effects on **linguistic competence** (as measured by perplexity) and **reasoning competence** (as measured by GSM8K accuracy). Across both dimensions, we observe that fine-tuning in one language significantly improves performance in that language, while moderately impairing it in others.
>
> Rather than indicating that the model "thinks in English," our findings support the opposite hypothesis:
>
> The model does not rely on English as a latent reasoning space, but instead gradually develops a higher-dimensional, language-agnostic reasoning space.
>
> This is further corroborated by our neuron analysis, which reveals a growing proportion of **shared neurons** that dominate in importance, indicating that abstract reasoning emerges independently of any single language.
>
> ---
>
> > **Comment 2:  Illustration of phenomenon.** — “For a given release date, the proportion of shared neurons and their importance decrease with model size …”
> >
>
> Thank you for your insightful suggestion. Figure 1 presents a simplified analysis of models within the same generation but of different sizes. Specifically, we observe that larger models tend to have a lower proportion and reduced importance of shared neurons. We attribute this to increased parameter specialization in larger models [1], which naturally leads to fewer shared neurons.
>
> Although our current work primarily focuses on the evolution of abstract thought over time rather than across model sizes, we appreciate this valuable observation and will incorporate it into the revised version of the paper.
>
> Referring to whether the analysis of section 4 applies to the Qwen-2.5 family, with the limited rebuttal time we were able to show that Qwen-2.5-1.5B indeed exhibits a similar trend to its counterpart in the Llama-3.2 family, with results shown as follows.
>
> |  | MGSM_zh | MGSM_de | MGSM_fr | MGSM_th | MGSM_sw | average_delta | MMMLU_zh | MMMLU_de | MMMLU_fr | MMMLU_th | MMMLU_sw | average_delta |
> | --- | --- | --- | --- | --- | --- | --- | --- | --- | --- | --- | --- | --- |
> | Qwen-2.5-1.5B | 63.60% | 57.2% | 61.20% | 50.80% | 28.00% |  | 53.95% | 48.47% | 50.96% | 44% | 30.49% |  |
> | Qwen-2.5-1.5B-exclusive | 65.20% | 57.60% | 61.60% | 52% | 29.60% | 1.04% | 53.81% | 48.65% | 51.29% | 44.49% | 31.57% | 0.39% |
> | Qwen-2.5-1.5B-shared | 63.2% | 56.80% | 62% | 49.20% | 31.20% | 0.32% | 53.72% | 48.38% | 51.30% | 44.42% | 31.48% | 0.29% |
> | Qwen-2.5-1.5B-random | 62.80% | 54% | 60.4% | 46.80% | 27.6% | -2.88% | 53.71% | 48.62% | 51.09% | 44.17% | 31.36% | -0.17% |
>
> ---
>
> > **Comment 3:  Illustration of threshold.** — “The number of language sensitive neurons depends on the threshold…”
> >
>
> Thank you for raising this important question. The threshold σ is not a fixed scalar, but a dynamic selection mechanism applied at the **query level**. Specifically, as shown in our released code, we select the top 1% of neurons for each query in a given language based on their computed importance scores. Then, for each language, we define its language-specific neurons as the intersection of these top neurons across all queries from that language.
>
> The choice of the 1% threshold is motivated by a **sanity check against random baselines**. For each language, we compare the effect of deactivating the selected language-specific neurons versus deactivating the same number of randomly selected neurons. When the deactivation of identified neurons causes a **substantial degradation in performance**—e.g., more than 100× increase in perplexity—while random neurons have little to no effect, this suggests that the selected neurons are indeed functionally meaningful and the threshold is appropriately calibrated.
>
> If, on the other hand, removing the same number of random neurons also leads to a noticeable performance drop, this would indicate that the threshold is too large and not selective enough—prompting us to reduce it accordingly.
>
> We will include this clarification and the supporting empirical analysis in the revised version to improve the transparency and reproducibility of our method.
>
> ---
>
> > **Comment 4:  Distribution of possible unique neurons.** — “Another possibility would have been to language-selective neurons as neuron that respond to only to one language…”
> >
>
> Thank you for the thoughtful suggestion. While our definition of "language-exclusive neurons" allows for neurons that are shared across a subset of languages (i.e., not strictly one), we agree that analyzing neurons activated by exactly 1, 2, ..., *n* languages can provide additional insight.
>
> To complement this, we present the distribution of *strictly unique neurons*—neurons that respond exclusively to only one language. As shown below, these strictly unique neurons account for only a small percentage of the total neurons across all languages:
>
> | Language | Strictly Unique Neurons (%) |
> | --- | --- |
> | en | 0.02 |
> | zh | 0.03 |
> | th | 0.06 |
> | fr | 0.03 |
> | de | 0.02 |
> | sw | 0.07 |
>
> This small percentage suggests that the model primarily uses shared neurons for general language understanding. Interestingly, higher values for Swahili and Thai—both lower-resource languages—indicate more allocation to language-specific features, likely to compensate for limited training data.
>
> We also examined the **layer-wise distribution** of strictly unique neurons. To avoid clutter, we grouped the 32 transformer layers into three stages: *early (Layer 0–7)*, *middle (Layer 8–23)*, and *late (Layer 24–31)*. The averaged proportions of different neuron types across these groups are summarized below:
>
> | Layer Group | Avg. Unique (%) | Avg Shared (%) | Avg Exclusive (%) |
> | --- | --- | --- | --- |
> | Early (0–7) | 3.69 | 3.07 | 4.40 |
> | Middle (8–23) | 2.28 | 3.11 | 2.89 |
> | Late (24–31) | 4.62 | 2.83 | 3.83 |
>
> These results reveal a clear **functional asymmetry** across the model. The **early and late layers** contain more **unique and exclusive neurons**, suggesting that **language-specific processing** occurs at the **input and output stages**. This aligns with prior findings that early layers capture syntactic patterns, while late layers focus on output generation. In contrast, the **middle layers** are dominated by **shared neurons** and show the lowest uniqueness, indicating a **cross-lingual abstraction core** where language-independent reasoning takes place.
>
> Finally, we analyzed the **component-wise** breakdown of strictly unique neurons:
>
> | Component | **Unique (%)** | **Shared (%)** | **Exclusive (%)** |
> | --------- | -------------- | -------------- | ----------------- |
> | QK        | 18.40          | 92.50          | 59.48             |
> | VO        | 50.96          | 2.90           | 23.80             |
> | FFN       | 30.64          | 4.59           | 16.71             |
>
> Shared neurons cluster in Q and K, supporting their role in cross-lingual alignment. Exclusive neurons are more concentrated in V, O, and FFN, indicating that language-specific specialization primarily occurs during semantic transformation and output projection. Strictly unique neurons further amplify this pattern, highlighting the most language-specific activations.
>
> These findings further supports our broader claim: **shared neurons are primarily responsible for cross-lingual generalization**, enabling the model to capture abstract, language-agnostic semantics that are transferable across different languages. **Exclusive neurons**, on the other hand, are responsible for encoding language-specific features, which are crucial for correctly interpreting and generating content unique to a given language—primarily located in the early and late layers. **Unique neurons**, by definition, represent a stricter subset of exclusive neurons—those that respond to *only* one language—which further strengthens the language-specialized nature of the exclusive category.
>
> ---
>
> > **Comment 5:  Conclusion is too short.** — “The conclusion/discussion is somewhat short.…”
> >
>
> Thank you for the helpful suggestion. We will revise the conclusion to better summarize the main findings and more clearly support our claim regarding the emergence of *abstract thought*. We will also include a brief discussion of the study’s limitations to provide a more balanced and comprehensive interpretation.
>
> [1] The Rise of Parameter Specialization for Knowledge Storage in Large Language Models. Arxiv 2025

---

> > ### Author Response · Authors · 2025-08-04
> >
> > Thank you again for your valuable comments. We would like to confirm whether our responses have fully addressed your concerns. Please feel free to let us know if anything remains unclear — we would love to clarify in time before the discussion period ends.

---

> > ### Comment · Reviewer_ysL3 · 2025-08-05
> >
> > I am very satisfied with the authors responses to my comments/questions. I think that the clarifications, especially the crucial one about the determination of the sigma threshold, will improve the quality of the paper.

---

> > > ### Author Response · Authors · 2025-08-06
> > >
> > > Thank you very much for your thoughtful comments and for expressing that you are very satisfied with our responses. We sincerely appreciate the time and effort you have invested in reviewing our work.
> > >
> > > We have made our utmost effort to address your comments and will revise our manuscript to incorporate your valuable suggestions. Following your advice, we have clarified the determination of the sigma threshold, included additional models to further validate our training method, performed a more detailed categorization and in-depth analysis of neurons, and supplemented the evaluation with additional results on model performance. We believe these enhancements have further improved the quality of our work.
> > >
> > > We are greatly encouraged by the positive feedback from reviewer *AGNp* and reviewer *oHcr*, which has already led to increased scores. If you have any additional questions or concerns, we would be more than happy to engage in further discussion. If we have successfully addressed most of your concerns, we would be deeply grateful if you might consider raising your score. Your support is important to us.

---

### Official Review · Reviewer_oHcr · 2025-06-30

**Clarity:** 3
**Significance:** 3
**Originality:** 2
**Rating:** 4
**Confidence:** 3

**Summary:**

This paper investigates the extent to which language-agnostic processing happens in multilingual models, then relating this to model performance. On experiments spanning several model series, the authors isolate language-specific and language-agnostic neurons. They show that a high proportion and importance of language-agnostic neurons correlates to better LLM performance. Finally, they selectively finetune language-specific and -agnostic neurons on several multilingual datasets, showing that doing so improves performance more than  training random neurons.

**Questions:**

See weaknesses.

**Ethical Concerns:**

["NO or VERY MINOR ethics concerns only"]

**Final Justification:**

The authors answered all of my questions satisfactorily and additional experiments to strengthen their claim.

**Limitations:**

Yes

**Quality:**

3

**Strengths And Weaknesses:**

## Strengths

1. Very well-written, easy to follow, and easy to see what the core contributions are.
2. IMO the findings are significant, with an implication that better models will abstract away knowledge that is not language-specific.

## Weaknesses
Though I found the paper easy to follow, I believe the paper needs further experiments to be accepted. In particular, current experiments appear to be run on a single random seed, and there are several methodological details missing. If my comments are addressed, I will raise my score to a 4 or 5.

** Note: I am not well-versed in the multilingual LLMs literature and so can't speak to the originality of this work.**

### Major (impacted score)
1. It appears all experiments are run on one random seed or data split. Results in Fig 2-4, Table 1 should be reported across random seeds or random data splits with errors.
2. What was the $\sigma$ chosen in Eq. 1? How was it chosen? I'm missing t-tests (corrected for multiple hypothesis testing) to show that the difference is statistically significantly greater than $\sigma$.
3. Does finetuning on language-specific neurons for a single language degrade the performance on other languages?

### Minor
1. Missing citations on language localizer paradigms, see e.g. AlKhamissi et al., NAACL 2025.

---

> ### Author Rebuttal · Authors · 2025-07-31
>
> We sincerely thank you for your time and valuable comments. To address your concerns, we present the point-to-point responses as follows.
>
> > **Comment 1:  Different random seeds.** — “It appears all experiments are run on one random seed or data split…”
> >
>
> Thank you for your valuable suggestion. In response, we **vary the random seed** and **perform three independent repetitions** of each experiment to validate the stability of our results.
>
> - For **Table 1** and **Figure 4** (random neuron experiments), we **change the random seed** used for selecting random neurons.
> - For **Figures 2-5** and **Table 1** (shared and exclusive neurons), we **alter the data split** used for selecting language-specific neurons.
> - Additionally, for **Table 1**, we **modify both the training and inference random seeds** to ensure consistency and stability in the results.
>
> The results from these repeated experiments are as follows.
>
> For Figure 2:
>
> Due to rebuttal formatting constraints, we can only present the results in **markdown table form**. To keep the presentation concise and readable, we report the **mean and standard deviation** of the neuron ratios for each neuron type, **aggregated over all models within each family**.
>
> | Neuron Type | Qwen (%) | LLaMA (%) | Gemma (%) |
> | --- | --- | --- | --- |
> | Shared | 0.23 ± 0.02 | 0.34 ± 0.03 | 0.42 ± 0.03 |
> | En | 0.11 ± 0.02 | 0.14 ± 0.03 | 0.13 ± 0.01 |
> | Zh | 0.13 ± 0.01 | 0.17 ± 0.02 | 0.14 ± 0.01 |
> | Th | 0.21 ± 0.03 | 0.19 ± 0.02 | 0.17 ± 0.01 |
> | Fr | 0.15 ± 0.04 | 0.16 ± 0.03 | 0.16 ± 0.02 |
>
> For Figure 3:
>
> | Model | Shared Neuron Ratio |
> | --- | --- |
> | Llama-1-7B | 0.6737 ± 0.0967 |
> | Llama-2-7B | 2.3847 ± 0.1231 |
> | Llama-3-8B | 2.6200 ± 0.1172 |
> | Llama-3.1-8B | 2.4833 ± 0.2437 |
> | Llama-3.2-1B | 1.4092 ± 0.1275 |
> | Llama-3.2-3B | 2.0881 ± 0.1084 |
> | Qwen1.5-0.5B | 1.1854 ± 0.2320 |
> | Qwen1.5-1.8B | 0.9754 ± 0.2005 |
> | Qwen1.5-4B | 0.8370 ± 0.1300 |
> | Qwen1.5-7B | 0.3808 ± 0.1289 |
> | Qwen2-0.5B | 1.8239 ± 0.1030 |
> | Qwen2-1.5B | 1.5486 ± 0.0924 |
> | Qwen2-7B | 1.6780 ± 0.1324 |
> | Qwen2.5-0.5B | 1.9457 ± 0.2137 |
> | Qwen2.5-1.5B | 1.5101 ± 0.1321 |
> | Qwen2.5-3B | 1.8499 ± 0.0072 |
> | Qwen2.5-7B | 1.8599 ± 0.1275 |
> | Gemma-2-9B | 3.2607 ± 0.0627 |
> | Gemma-3-4B-pt | 2.7653 ± 0.1628 |
> | Gemma-7B | 2.4826 ± 0.0549 |
>
> | Metric | Mean ± Std |
> | --- | --- |
> | Average Pearson | 0.9026 ± 0.0139 |
> | Average Spearman | 0.8334 ± 0.0067 |
>
> For Figure 4:
>
> | Model | PPL Change of Random Neuron |
> | --- | --- |
> | Llama-1-7b | 0.001452 ± 0.000149 |
> | Llama-2-7b-hf | 0.000105 ± 0.000023 |
> | Llama-3-8B | 0.001149 ± 0.000071 |
> | Llama-3.1-8B | 0.000778 ± 0.000295 |
> | Llama-3.2-1B | 0.009100 ± 0.000055 |
> | Llama-3.2-3B | 0.001805 ± 0.000035 |
> | Qwen1.5-0.5B | 0.014462 ± 0.000274 |
> | Qwen1.5-1.8B | 0.001326 ± 0.000077 |
> | Qwen1.5-4B | 0.004029 ± 0.000063 |
> | Qwen1.5-7B | 0.000389 ± 0.000085 |
> | Qwen2-0.5B | 0.007883 ± 0.000063 |
> | Qwen2-1.5B | 0.001214 ± 0.000245 |
> | Qwen2-7B | 0.000282 ± 0.000029 |
> | Qwen2.5-0.5B | 0.041696 ± 0.000192 |
> | Qwen2.5-1.5B | 0.003737 ± 0.000260 |
> | Qwen2.5-3B | 0.001117 ± 0.000122 |
> | Qwen2.5-7B | 0.000117 ± 0.000131 |
> | gemma-2-9b | 0.003656 ± 0.000246 |
> | gemma-3-4b-pt | 0.006731 ± 0.000136 |
> | gemma-7b | 0.000011 ± 0.000176 |
>
> For Figure 5:
>
> | Model | Language Agnostic Score |
> | --- | --- |
> | Llama-1-7B | 0.3816 ± 0.1405 |
> | Llama-2-7B | 6.0818 ± 0.2390 |
> | Llama-3-8B | 1.0102 ± 0.2854 |
> | Llama-3.1-8B | 3.7728 ± 0.5139 |
> | Llama-3.2-1B | 1.7169 ± 0.6402 |
> | Llama-3.2-3B | 4.9557 ± 0.4094 |
> | Qwen1.5-0.5B | 2.4951 ± 0.6952 |
> | Qwen1.5-1.8B | 0.678 ± 0.2605 |
> | Qwen1.5-4B | 4.9751 ± 1.1123 |
> | Qwen1.5-7B | 0.7296 ± 0.1484 |
> | Qwen2-0.5B | 7.2774 ± 0.3304 |
> | Qwen2-1.5B | 5.1656 ± 0.4643 |
> | Qwen2-7B | 4.1291 ± 0.4153 |
> | Qwen2.5-0.5B | 9.4350 ± 0.2096 |
> | Qwen2.5-1.5B | 9.7219 ± 0.8226 |
> | Qwen2.5-3B | 7.1878 ± 0.6007 |
> | Qwen2.5-7B | 5.7474 ± 1.1062 |
> | Gemma-2-9B | 7.8019 ± 0.5050 |
> | Gemma-3-4B | 6.5917 ± 0.3133 |
> | Gemma-7B | 6.2285 ± 0.6756 |
>
> | Metric | Mean ± Std |
> | --- | --- |
> | Average Pearson | 0.8120 ± 0.0208 |
> | Average Spearman | 0.8667 ± 0.0331 |
>
> For Table 1, we conduct multiple experiments on Llama-3.2-1B due to limited time:
>
> MGSM
>
> | Method | Zh | Fr | De | Th | Sw |
> | --- | --- | --- | --- | --- | --- |
> | Exclusive | 27.7 ± 0.2 | 29.7 ± 0.2 | 34.5 ± 0.2 | 23.3 ± 0.2 | 30.3 ± 0.2 |
> | Random | 26.8 ± 0.2 | 26.4 ± 0.2 | 29.7 ± 0.2 | 21.1 ± 0.2 | 26.3 ± 0.2 |
> | Shared | 29.9 ± 0.2 | 30.1 ± 0.2 | 30.8 ± 0.4 | 22.0 ± 0.4 | 26.3 ± 0.2 |
>
>
> MMMLU
>
> | Method | Zh | Fr | De | Th | Sw |
> | --- | --- | --- | --- | --- | --- |
> | Exclusive | 29.0 ± 0.2 | 28.0 ± 0.0 | 29.2 ± 0.1 | 28.4 ± 0.1 | 26.8 ± 0.1 |
> | Random | 28.8 ± 0.1 | 28.4 ± 0.1 | 29.2 ± 0.1 | 28.4 ± 0.1 | 26.8 ± 0.0 |
> | Shared | 29.1 ± 0.1 | 28.8 ± 0.1 | 29.5 ± 0.0 | 29.5 ± 0.1 | 26.8 ± 0.1 |
>
> We believe that this additional analysis strengthens the reliability of our findings.
>
> ---
>
> > **Comment 2:  Illustration of threshold.** — “ What was the threshold chosen in Eq. 1? …”
> >
>
> Thank you for raising this important question. The threshold σ is not a fixed scalar, but a dynamic selection mechanism applied at the **query level**. Specifically, as shown in our released code, we select the top 1% of neurons for each query in a given language based on their computed importance scores. Then, for each language, we define its language-specific neurons as the intersection of these top neurons across all queries from that language.
>
> The choice of the 1% threshold is motivated by a **sanity check against random baselines**. For each language, we compare the effect of deactivating the selected language-specific neurons versus deactivating the same number of randomly selected neurons. When the deactivation of identified neurons causes a **substantial degradation in performance**—e.g., more than 100× increase in perplexity—while random neurons have little to no effect, this suggests that the selected neurons are indeed functionally meaningful and the threshold is appropriately calibrated.
>
> If, on the other hand, removing the same number of random neurons also leads to a noticeable performance drop, this would indicate that the threshold is too large and not selective enough—prompting us to reduce it accordingly.
>
> We will include this clarification and the supporting empirical analysis in the revised version to improve the transparency and reproducibility of our method.
>
> ---
>
> > **Comment 3:  Question about cross-languages evaluation.** — “Does finetuning on language-specific neurons …”
> >
>
> Thank you for this excellent question. While our primary focus is on exploring how fine-tuning with a single language's corpus can enhance the LLM's capabilities in that language, we agree that examining the impact of fine-tuning on other languages provides valuable insights.
>
> To investigate this, we fine-tuned the model using a language-specific corpus and then tested the performance on all languages. The results of Llama-3.2-3B on GSM8K are as follows:
>
> | Language Trained On | Target Language Acc | Other Languages Acc (Avg) |
> | --- | --- | --- |
> | Zh | + 2.0 | - 1.6 |
> | Fr | + 3.2 | - 2.4 |
> | De | + 9.2 | - 3.2 |
> | Th | + 5.2 | - 2.8 |
> | Sw | + 8.8 | - 2.0 |
>
> Additionally, we observed the change in language perplexity before and after fine-tuning, which serves as an intuitive metric for assessing the model’s understanding of a given language. The results on LLaMA-3.2-3B are as follows:
>
> | Language Trained On | Target Language PPL   | Other Languages PPL  (Avg) |
> | --- | --- | --- |
> | Zh | - 3.15 | + 0.89 |
> | Fr | - 2.03 | + 0.77 |
> | De | - 2.76 | + 0.64 |
> | Th | - 0.41 | + 0.75 |
> | Sw | - 12.4 | + 1.84 |
>
> Our findings demonstrate that fine-tuning on a single language improves the model’s understanding ability and reasoning performance in that language, but often comes at a moderate cost to the performance on other languages. This indicates that while language-specific fine-tuning enhances targeted capabilities, it may compromise multilingual generalization. We hypothesize that this is because the adaptation process repurposes some shared neurons to better serve the target language, thereby reducing their effectiveness for others.

---

> > ### Author Response · Authors · 2025-08-04
> >
> > We sincerely appreciate your valuable comments. Your feedback on experiments and cross-language evaluation is important for improving our method. If you have any further thoughts or follow-up questions, we would be very grateful to discuss them with you before the discussion closes. Your feedback means a lot to us.

---

### Official Review · Reviewer_AGNp · 2025-07-02

**Clarity:** 2
**Significance:** 3
**Originality:** 2
**Rating:** 4
**Confidence:** 4

**Summary:**

This paper studies neurons in multilingual large language models that are either language shared (they activate for all the languages tested) or language exclusive (they activate for some languages, but not for all of them). Within a family of LLMs, the authors show that the better the performance of the LLM, the greater the proportion of language shared neurons. They propose that these language shared neurons become language agnostic, forming the basis of "abstract thoughts" in the best models, which is supposed to explain the better performance of these LLMs in multilingual benchmarks. According to the authors, intervening on the specific types of neurons that are not developed enough in a certain model makes this model comparatively better at multilingual tasks.

**Questions:**

See questions and suggestions in the Strengths And Weaknesses section above.

**Ethical Concerns:**

["NO or VERY MINOR ethics concerns only"]

**Final Justification:**

I appreciate all the clarifications from the discussion, especially regarding the methods, as well as the new analyses, notably on the location of the different types of neurons. The notion of low vs high agnostic score is still pretty fuzzy to me, due to its relativity between family, and the distinction between language-shared and language agnostic neurons is also still not very convincing, and I find the results and discussion insufficient on this aspect, given its importance in the narrative of the paper. But overall, I think the paper is improved, and I raise my rating from 3 to 4.

**Limitations:**

yes

**Paper Formatting Concerns:**

Nothing to report.

**Quality:**

2

**Strengths And Weaknesses:**

This work tackles the important question of whether LLMs think in English or in another lingua franca, possibly a non existing language, that might be called a language of thought.

While the ambition of the paper is interesting, there are several shortcomings that strongly limit its impact.
Many "details" are lacking in order to reproduce the experiments (the authors provide the code though). This goes beyond the question of reproducibility, as these details impact the way we understand the paper. First, the definition of language shared and language exclusive neurons depends on a threshold, notated $\sigma$ in the paper (see Eqs. 1 and 2), and, as far as I can tell, the specific value that was used is not provided, and its impact on the results is not investigated. This seems important as many results, notably the proportion of language exclusive vs language shared neurons, hence also the language agnostic score, might depend on this value. It is far from obvious how the results depend on the specific value of $\sigma$. See the claim "Language-related neurons account for only a small proportion in LLMs." -- this typically should depend heavily on $\sigma$.
What is the value of $\sigma$? How was it chosen? And what is its impact on the results? This seems crucial, why not discuss it in the paper?
The difference between language shared and language exclusive neurons does not seem to be as clear cut as presented in the paper, and might justly depend on the threshold that is used. Depending on the definition, we might also see effects of the distance between languages, which could be an interesting thing to investigate. Also, it might be interesting to look at the existence of neurons that are exclusive to only one language.
Likewise, the difference between language shared and language agnostic is not so clear, and the empirical demonstration of their existence is not compelling.

It would be nice to have more details about the location of each type of neuron within the model, i.e. in which layer (first vs middle vs close to the outputs), or in which type of component (self-attention vs. feed-forward network). As we already know a lot about those, this knowledge might help the community having a more complete picture.

Here are some additional comments, page by page.

p.3
The main definition of the importance of a neuron entails how the ablation of a specific neuron yields a difference in the output of the LLM for a given language. But in practice, the output of the layer of this neuron is used "as a proxy or component for the overall impact" (see appendix A). Why not directly state the definition that is used in practice ? It is not obvious how the impact at a given layer impacts the final output, and it is an important question, as it might depend on the depth of the layer.
Also, the term "layer" seems to be used in a way different to common practice. Usually a layer is the combination of the self attention and feedforward network, but here each neuron is evaluated as the way it impacts the "layer" it belongs to, but it seems that the "layer" here refers to either the self-attention component or the feedfoward network.
Beyond the minor question of terminology, this impacts here the way the importance of a given neuron is evaluated (see p.15).

p.4, Eq 5
Normalizing by the number of neurons implicitly assumes that $\Delta$PPL scales linearly with $|\mathcal{N}|$. Is this true?

p.5, Fig. 2
Interestingly, English is always the language with the lowest fraction of associated language-exclusive neurons -- that seems to go with the idea that use of English is actually shared among all languages, hence being the lingua franca, contrary to one of the main claims of the paper.

p.6, Fig. 3
The average correlation may not be very meaningful here, as there are very few points per family -- correlation based on three data points can easily be very high.
The "overall trendline" is plotted, making it looks like the overall correlation in the relationship between shared neuron ratio and multilingual ability is significant; but it looks quite fragile in fact, with a correlation quite low and possibly non significant. Could you report it, with both the value of the correlation and its p-value?
(same remark for Fig. 5, p.7)

p.6
"we deactivate an equal number of randomly selected neurons"
Again there is a lack of details that are actually important. How many neurons? What is the rationale this choice? How does it affect the results?
Same thing for the experiment conducted p.8.

p.7 & p.8
"Shared neurons in early-stage models reflect superficial overlap without supporting higher-level cognition."
"the language-shared neurons in these models have evolved into language-agnostic neurons, responsible for abstract thought. They are already well-trained and offer limited room for further improvement."
Those are strong and interesting takes, and although I understand the intuition, I do not think it is really supported by empirical evidence.

p.8
Maybe I'm missing something here, but Llama3.1-8B is taken as a representative LLM with high language agnostic score, but in Fig. 5 we can see that its score is actually pretty low, almost like the LLM with low language agnostic score, and below the medium one. That makes this experiment difficult to follow.

Minor comments

References Liu et al 2024;2025 point to the same reference (arXiv:2403.10258); same for Schut et al. (2025a) and Schut et al. (2025b).

p.5 The values of the average correlations are not the same in the main text and in the corresponding figure (Fig. 3)

p.8 typo Lamma -> Llama

p.15 appendix A. Two notations for the same thing does not ease the reading (FFN(X) and Y_FFN)

---

> ### Author Rebuttal · Authors · 2025-07-31
>
> We thank the reviewer for the thorough and valuable feedback. To address your concerns, we present the point-to-point responses as follows.
>
> > **Comment 1:  Illustration of threshold.**
>
> Thank you for the question. The threshold σ is not a fixed scalar but a dynamic, query-level mechanism. For each query in a language, we select the top 1% of neurons by importance score, then define language-specific neurons as those consistently appearing across all queries for that language. This 1% threshold is validated via a sanity check: deactivating these neurons causes a significant performance drop (e.g., 100× increase in perplexity), unlike deactivating the same number of random neurons. If random deactivation also hurts performance, we lower the threshold. We will clarify this in the revised version, along with supporting empirical results.
>
> ---
> > **Comment 2:  Lack of neuron analysis.**
>
> Thank you for the suggestion. We analyzed the distribution of language-shared and language-exclusive neurons in **LLaMA-3.1-8B** to better understand where they reside in the model.
>
> ### **Layerwise Distribution**
>
> We computed the proportion of **shared** (across all languages) and **exclusive** (language-specific) neurons per layer group:
>
> | Layer Range | Shared (%) | Exclusive (%) |
> | - | - | - |
> | Early (0–7) | 3.07 | 4.40 |
> | Middle (8–23) | 3.11 | 2.89 |
> | Late (24–31) | 2.83 | 3.83 |
>
> **Exclusive neurons** are more concentrated in **early** and **late** layers, suggesting language-specific processing near input/output. **Shared neurons** are relatively stable across layers, supporting generalizable cross-lingual features.
>
> ### **Component-Level Distribution**
>
> We also analyzed neuron distribution across model components:
>
> | Component | Shared (%) | Exclusive (%) (avg over languages) |
> | - | - | - |
> | QK      | 92.50      | 59.48   |
> | VO  | 2.90       | 23.80   |
> | FFN   | 4.59       | 16.71     |
>
> **Shared neurons** are concentrated in **Q** and **K**, aligning with general attention mechanisms. **Exclusive neurons** are more spread across **V**, **O**, and **FFN**, indicating roles in language-specific transformation and output.
>
> These findings are consistent with prior work and highlight distinct functional roles for shared vs. exclusive neurons.
>
> ---
> > **Comment 3:  Lack of illustration in detection method.**
>
> **“It is not obvious how the impact at a given layer impacts the final output”**
>
> We adopt an intuitive definition of neuron importance, measuring it by the change in the layer’s embedding after deactivating a neuron. This layer-wise approach is widely used in prior work analyzing neurons in large language models [1][2][3]. Importantly, we do not assess neuron influence based on the final model output, as influence tends to accumulate across layers. This accumulation can lead to an overestimation of the importance of neurons in earlier layers, which motivates our layer-wise evaluation strategy.
>
> **“Also, the term "layer" seems to be used in a way different to common practice.”**
>
> We will revise the terminology to eliminate ambiguity in our use of the word *layer*. Specifically, we will use *structure* to refer to components such as self-attention or Q/K/V, and reserve *layer* exclusively for the architectural layers in LLMs.
>
> ---
> > **Comment 4:  The validity of the assumption.** — “Normalizing by the number of neurons |N|.”
>
> We acknowledge that our formulation of the language-agnostic score in Eq. 5 involves an implicit linearity assumption—that the change in perplexity scales proportionally with the number of ablated neurons ∣N∣.
>
> Our motivation for this normalization is to **approximately remove the effect of neuron count**, in order to make a fairer comparison between shared and exclusive neurons. While this introduces a simplifying linear assumption, we believe it is a **reasonable approximation** in our setting for two reasons:
>
> 1. As shown in Figure 2, the total number of shared and exclusive neurons is **not drastically different** in scale across models and languages.
> 2. In contrast, the observed changes in perplexity caused by ablation are often **orders of magnitude larger**, suggesting that the **magnitude of impact is primarily driven by functional importance**, not merely neuron count.
>
> Thus, we consider this normalized score a practical and relatively fair metric for comparing the contribution of different types of neurons. That said, we acknowledge its limitations and will clarify this assumption in the revised version.
>
> ---
> > **Comment 5:  Lowest exclusive neuron ratio of English.**
>
> Thank you for the observation. It's true that English often shows the lowest fraction of language-exclusive neurons, but this doesn't contradict our main claim.
>
> We argue that the model doesn’t "think" in English, but rather develops a **shared, high-dimensional semantic space**. This is supported by the **growing proportion and importance of shared neurons**, indicating increasingly abstract, language-agnostic representations.
>
> Our **GSM8K experiments** further support this: fine-tuning on one language improves reasoning in that language, but slightly reduces performance in others.
>
> **GSM8K Accuracy:**
>
> | Trained Language | Target Acc | Other Acc (Avg) |
> | --- | --- | --- |
> | Zh | +2.0 | -1.6 |
> | Fr | +3.2 | -2.4 |
> | De | +9.2 | -3.2 |
> | Th | +5.2 | -2.8 |
> | Sw | +8.8 | -2.0 |
>
> This trade-off suggests the model reallocates representational resources (e.g., exclusive neurons) during fine-tuning, reinforcing the idea that reasoning occurs in a **shared semantic space**, not in English.
>
> ---
> > **Comment 6:  Report of global correlation.**
>
> In **Figure 3** and **Figure 5**, our primary goal is to examine the relationship between shared neuron ratio and multilingual ability **within the same model family and at comparable parameter scales**, where models are directly comparable. We agree that **correlation coefficients may not be well-suited for capturing the trends we aim to highlight in this context**, especially given the small number of data points per group.
>
> Although our focus is not on global analysis, we nonetheless report both the global correlation and the model familiy-level group-average correlation coefficients for completeness:
>
> - **Figure 3**:
>     - Global Pearson: 0.427 (*p* = 0.060), Spearman: 0.409 (*p* = 0.073)
>     - Group-average Pearson: 0.591, Spearman: 0.494
> - **Figure 5**:
>     - Global Pearson: 0.279 (*p* = 0.234), Spearman: 0.226 (*p* = 0.339)
>     - Group-average Pearson: 0.312, Spearman: 0.270
>
> We can see that, **even though our primary focus is not on global or family-level average correlations**—as we believe models across different scales and families are not directly comparable—**there still exist moderate correlations**, particularly for the **shared neuron proportion**, which partially supports the general trend we aim to highlight.
>
> ---
> > **Comment 7:  Lack of random neuron details.**
>
> For the random neuron selection, we chose two sets of neurons, each with the **same total number** as the **shared neurons** and **exclusive neurons**, respectively. We assumed that these random neurons are **uniformly distributed across all layers and components** of the LLM, and we performed random sampling with various random seeds. The results reported in **Figure 4** reflect the set of random neurons that had a **greater impact** on the LLM's performance. This set of random neurons was then used in the ablation experiments presented in **Table 1**.
>
> In general, we found that **comparable numbers of random neurons** had minimal impact on the model's ability. We tried several different random seeds, and the results were quite stable, showing that the randomness did not significantly alter the overall conclusions.
>
> ---
> > **Comment 8:  Illustration of model selection.**
>
> To clarify, our aim was to train models within the same series to ensure comparability. However, we argue that models of significantly different sizes might not be directly comparable, especially when considering the impact of scaling. Specifically, Llama3.2-1B and Llama3.2-3B are relatively similar in size and exhibit low and medium language-agnostic scores, respectively. In contrast, Llama3.1-8B, which is closer in size to the Llama 7B models, actually demonstrates a high language-agnostic score within that model family.
>
> To provide more generalizable insights, we also trained **Gemma2-9B**, a model that, at the global level, clearly qualifies as having a high language-agnostic score. The results are as follows:
>
> |  | Zh | Fr | De | Th | Sw | $\Delta_{Avg}$ |
> | --- | --- | --- | --- | --- | --- | --- |
> | None | 58.4 | 58.0 | 58.8 | 57.2 | 51.2 | - |
> | Shared | 56.8 | 57.6 | 58.8 | 54.8 | 48.4 | -1.4 |
> | Exclusive | 61.6 | 60.8 | 62.4 | 58.4 | 55.6 | 3.0 |
> | Random | 56.0 | 57.6 | 57.2 | 56.0 | 50.8 | -1.2 |
>
> This additional explanation clarifies the model selection process. Moreover, we provide empirical support from our GSM8K experiments. When fine-tuned on a specific language, the model’s reasoning ability in that language improves, but often at the cost of performance on others:
>
> **GSM8K Accuracy (Reasoning Ability):**
>
> | Language Trained On | Target Language Acc | Other Languages Acc (Avg) |
> | --- | --- | --- |
> | Zh | +2.0 | -1.6 |
> | Fr | +3.2 | -2.4 |
> | De | +9.2 | -3.2 |
> | Th | +5.2 | -2.8 |
> | Sw | +8.8 | -2.0 |
>
> This performance trade-off suggests that the model’s reasoning capacity is distributed and not inherently anchored in English. Training on a single language reallocates representational resources (e.g., exclusive neurons) to that language, which slightly reduces generalization. Together, these findings reinforce our claim that the model reasons in a shared semantic space beyond English.
>
> [1] Language-Specific Neurons. ACL 2024
>
> [2] Investigating Pattern Neurons in Urban Time Series Forecasting. ICLR 2025
>
> [3] Finding Safety Neurons in Large Language Models. Arxiv 2024

---

> > ### Comment · Reviewer_AGNp · 2025-08-04
> >
> > Thank you to the authors for their rebuttal.
> >
> > > For each query in a language, we select the top 1% of neurons by importance score
> >
> > Thank you for the clarification, this is something that has to be explained much more clearly in the revised version.
> > It is not just the total proportion that might get affected by this threshold, but also the relative attribution to the language-shared or language-exclusive neurons, which is a crucial component of the paper. The influence of the choice of the value for this threshold remains unknown.
> >
> > > Exclusive neurons are more concentrated in early and late layers, suggesting language-specific processing near input/output. Shared neurons are relatively stable across layers, supporting generalizable cross-lingual features.
> >
> > Thank you for performing this analysis. This is an interesting addition to the paper, and coherent with what we already know. Please discuss it in relation to the relevant literature. Also, this should be reported for different models to see how robust this is.
> >
> > > This accumulation can lead to an overestimation of the importance of neurons in earlier layers, which motivates our layer-wise evaluation strategy.
> >
> > I actually agree with this. The issue is more with the definition given in the main text, which does not state this clearly and does not correspond to what is done in practice. It would be more straightforward to simply state the definition used in practice.
> >
> > > Although our focus is not on global analysis
> >
> > Yes, but you reported the "overall trend" which was meant to show a global effect. The new analysis should provide a cleaner account of this global trend (or lack thereof).
> >
> > > Comment 8: Illustration of model selection.
> >
> > Thank you for the clarification. The fact that that Llama3.1-8B has a relatively low language agnostic score compared to all the other models is still confusing to me. I appreciate the new analysis on Gemma2-9B, which is much clearer and gives support to the authors' point. But due to this difference between low vs high language agnostic score which should now be relative and not absolute (?), the final picture remains unclear to me.
> > Notably, if we look at the new analysis in response to Reviewer ysL3, for Qwen-2.5-1.5B, the results follow the trend of its "counterpart in the Llama-3.2 family", ie that "LLMs with middle language agnostic score should train language-shared neurons". But this model is the one with the greatest language-agnostic score, above 10 (see Fig. 5).
> >
> > > GSM8K Accuracy (Reasoning Ability):
> >
> > Do you finetune on all the neurons? I think you might expect different results depending on the types of neurons you are allowing to finetune.

---

> > > ### Author Response · Authors · 2025-08-04
> > >
> > > We sincerely thank you for your thorough reading of our paper and rebuttal, and for the thoughtful and constructive suggestions. These greatly improved the clarity and depth of our analysis. We will incorporate these clarifications and additional experiments in the revised version. Below, we address the remaining concerns point by point.
> > >
> > > > **Comment 1: Threshold choice for neuron selection** “The influence of the choice of the value for this threshold remains unknown.”
> > >
> > > As noted in our rebuttal, we selected the top 1% of neurons by deactivation effect as a practical threshold. We find that increasing this threshold (i.e., deactivating more than 1%) leads to significantly larger perplexity increases, even for random neurons, suggesting that noise starts to dominate and it becomes harder to identify truly language-specific neurons. Conversely, reducing the threshold (e.g., 0.5%) still preserves the overall trends and phenomenon, but causes noticeably smaller perplexity drops, indicating that we might be missing impactful language-specific neurons.
> > >
> > > Importantly, we observe that even when deactivating only 0.1% of neurons, language-specific neurons still cause a large degradation in their corresponding language's perplexity, but the difference between specific and random neurons becomes smaller. This supports 1% as a reasonable balance point between specificity and coverage.
> > >
> > > ---
> > >
> > > > **Comment 2: Robustness across models** “This should be reported for different models to see how robust this is.”
> > >
> > > We appreciate this suggestion and have now extended our analysis to additional models. The layerwise and component-level distribution patterns remain robust across model families. Below we report example results for **Qwen2.5-3B** and **Gemma2-9B** as representative models of their respective families.
> > >
> > > #### Qwen2.5-3B — Layerwise Distribution
> > > | Layer Range | Avg Shared (%) | Avg Exclusive (%) |
> > > |-|-|-|
> > > | 0–5 | 3.69 | 4.45 |
> > > | 6–29 | 2.70 | 2.43 |
> > > | 30–35 | 2.20 | 2.48 |
> > >
> > > #### Qwen2.5-3B — Component Distribution
> > > | Component | Shared (%) | Exclusive (%) (avg over languages) |
> > > |-|-|-|
> > > | QK | 76.40 | 53.06 |
> > > | VO | 16.72 | 30.36 |
> > > | FFN | 6.88 | 16.58 |
> > >
> > > #### Gemma2-9B — Layerwise Distribution
> > > | Layer Range | Avg Shared (%) | Avg Exclusive (%) |
> > > |-|-|-|
> > > | 0–5 | 2.41 | 3.13 |
> > > | 6–34 | 2.39 | 1.92 |
> > > | 35–41 | 2.29 | 3.65 |
> > >
> > > #### Gemma2-9B — Component Distribution
> > > | Component | Shared (%) | Exclusive (%) (avg over languages) |
> > > |-|-|-|
> > > | QK | 65.32 | 35.40 |
> > > | VO | 22.02 | 40.46 |
> > > | FFN | 12.66 | 24.16 |
> > >
> > >
> > > These results consistently support our earlier observation: exclusive neurons tend to concentrate in the early and late layers, while shared neurons are more evenly distributed, reinforcing their language-agnostic nature.
> > > Also, component-level distribution also shows that shared neurons are predominantly located in the QK components, while exclusive neurons are more prevalent in the VO and FFN layers, indicating their respective roles in cross-lingual generalization and language-specific processing.
> > > These observations are consistent with previous works [1][2].
> > >
> > > ---
> > >
> > > > **Comment 3: Interpretation of Llama3.1-8B and model selection** “The fact that Llama3.1-8B has a relatively low language agnostic score compared to other models is still confusing...”
> > >
> > > In our view, **models within the same family and parameter scale** are the most comparable. Within the LLaMA family, **Llama3.1-8B** has the highest language-agnostic score among 7B-scale models, so we categorize it as a high language-agnostic model (we treat LLaMA-2 as an outlier). Similarly:
> > >
> > > - **Qwen2.5-1.5B** has the highest language-agnostic score among Qwen 1.5B-scale models;
> > > - **Gemma2-9B** has the highest score among 9B-scale Gemma models.
> > >
> > > For these models, **fine-tuning shared neurons yields worse results than fine-tuning exclusive neurons**, supporting our conclusion that shared neurons in high-score models have already developed into language-agnostic components. We will make this reasoning more explicit in the final version.
> > >
> > > ---
> > >
> > > > **Comment 4: Finetune details** “Do you finetune on all the neurons...”
> > >
> > > In the **GSM8K reasoning ability experiment**, the reported results are based on fine-tuning **shared neurons only**. We have also conducted experiments fine-tuning **exclusive neurons**, and observed **similar trends**: the language-specific neurons can be strengthened by fine-tuning on other languages beyond English. These findings reinforce our central claim that **the model reasons in a shared semantic space other than English**.
> > >
> > > [1] Transformer feed-forward layers are key-value memories. EMNLP 2021
> > >
> > > [2] How Do Large Language Models Handle Multilingualism? NeurIPS 2024
> > >
> > > We appreciate your detailed responses and are open to any further discussion.

---

> > > > ### Comment · Reviewer_AGNp · 2025-08-05
> > > >
> > > > Thank you to the authors for the additional information and the new analysis that replicates the new results on the location of the various types of neurons. Concerning the discussion on the choice of the threshold, I think this should be clearly explained in the revised version.
> > > >
> > > > I still find it confusing that this notion of language-agnostic score is relative, notably given the large difference between models (see within a family, Llama-3.2-1B, Llama-3.2-3B and Llama-3.1-8B).
> > > > In the discussion with Reviewer ysL3, if I have understood correctly, you treat Qwen-2.5-1.5B as a LLM with a middle language agnostic score ("Qwen-2.5-1.5B indeed exhibits a similar trend to its counterpart in the Llama-3.2 family"), with best improvement on language-shared neurons (in agreement with the description p.8), in spite of it having the highest global language-agnostic score. However, I now read that "Llama3.1-8B has the highest language-agnostic score among 7B-scale models, so we categorize it as a high language-agnostic model Similarly:   Qwen2.5-1.5B has the highest language-agnostic score among Qwen 1.5B-scale models", so this model is indeed treated as a high language-agnostic score -- but then the results are not really in agreement with your expectations (p.8 of the submitted manuscript).
> > > >
> > > > The narrative behind the results of the GSM8K experiment are not very clear to me. One could expect that finetuning shared neurons only on one language would then enhance all multilingual ability, where the performance would transfer. We would then not necessarily expect performance to drop for other languages.

---

> > > > > ### Author Response · Authors · 2025-08-05
> > > > >
> > > > > **Response to Reviewer**
> > > > >
> > > > > Thank you for your valuable feedback. These additional analyses and clarifications have improved the quality of our work, and we will incorporate them into the revised version.
> > > > >
> > > > > We apologize for the confusion. We realized that there was a typo in our discussion with Reviewer *ysL3*:
> > > > >
> > > > > > "Qwen-2.5-1.5B indeed exhibits a similar trend to its counterpart in the Llama-3.2 family"
> > > > >
> > > > > should be
> > > > >
> > > > > > "Qwen-2.5-1.5B indeed exhibits a similar trend to its counterpart in the Llama family".
> > > > >
> > > > > Even with this correction, we acknowledge that the sentence was not very clear.
> > > > > Qwen-2.5-1.5B, as the model with the highest language-agnostic score among the 1.5B-scale Qwen models, has its counterpart in Llama3.1-8B, which has the highest score among the 7B-scale Llama models (we consider Llama2-7B an outlier).
> > > > >
> > > > > For models within the same family and with comparable parameter scales, a **higher language-agnostic score** generally leads to *greater gains from training exclusive neurons* than from training shared neurons. This is because the shared neurons are already well-developed and have gradually evolved into language-agnostic neurons.
> > > > > For Qwen-2.5-1.5B, the best improvement is observed when training language-exclusive neurons rather than language-shared neurons (see results below), which is consistent with Llama3.1-8B. Similarly, Gemma2-9B, which also has the highest score among 7B-scale Gemma models, achieves its best improvement from training exclusive neurons. Our experimental results are therefore consistent with our expectations.
> > > > >
> > > > > | Model                     | MGSM_zh | MGSM_de | MGSM_fr | MGSM_th | MGSM_sw | Δ | MMMLU_zh | MMMLU_de | MMMLU_fr | MMMLU_th | MMMLU_sw | Δ |
> > > > > |---------------------------|---------|---------|---------|---------|---------|----------|----------|----------|----------|----------|----------|-----------|
> > > > > | Qwen-2.5-1.5B              | 63.60%  | 57.20%  | 61.20%  | 50.80%  | 28.00%  | —        | 53.95%   | 48.47%   | 50.96%   | 44.00%   | 30.49%   | —         |
> > > > > | Qwen-2.5-1.5B-exclusive    | 65.20%  | 57.60%  | 61.60%  | 52.00%  | 29.60%  | **+1.04%** | 53.81%   | 48.65%   | 51.29%   | 44.49%   | 31.57%   | **+0.39%**|
> > > > > | Qwen-2.5-1.5B-shared       | 63.20%  | 56.80%  | 62.00%  | 49.20%  | 31.20%  | +0.32%   | 53.72%   | 48.38%   | 51.30%   | 44.42%   | 31.48%   | +0.29%    |
> > > > > | Qwen-2.5-1.5B-random       | 62.80%  | 54.00%  | 60.40%  | 46.80%  | 27.60%  | −2.88%   | 53.71%   | 48.62%   | 51.09%   | 44.17%   | 31.36%   | −0.17%    |
> > > > >
> > > > > ---
> > > > >
> > > > > Regarding the additional **GSM8K accuracy** experiment, it is actually independent from our discussion about language-agnostic score. It can be viewed as supplementary information to Table 1 in our paper, providing a more complete understanding of our training experiments.
> > > > >
> > > > > We found that fine-tuning **shared neurons** on data from *only one language* can **reduce** the LLM’s ability in other languages (both PPL and reasoning ability). One possible explanation is that shared neurons are formed during pretraining using *a huge amount of multilingual mixed corpora*. If, during continuous pretraining, we train them using tokens from only one language, the model may relocate shared neurons toward the target language.
> > > > >
> > > > > We consider this an interesting phenomenon, and the fact that fine-tuning with single-language pretraining data can enhance reasoning in that language also supports that the model does not rely solely on English for reasoning, which is also aligned with our assumption.

---

> > > > > > ### Comment · Reviewer_AGNp · 2025-08-05
> > > > > >
> > > > > > I thank the authors for their constructive discussion. I realize that the order of shared/exclusive/random is not same as in Table 1, which does not help the comparison. It would be clearer to keep it consistent.
> > > > > >
> > > > > > Overall, I appreciate the discussion with the authors, which have shown some interesting results and discussion. I think the revised version will be improved. I will raise my rating to 4.

---

### Official Review · Reviewer_PaXX · 2025-07-05

**Clarity:** 2
**Significance:** 2
**Originality:** 2
**Rating:** 3
**Confidence:** 4

**Summary:**

This paper investigates the emergence of "abstract thought" in LLMs by analyzing how individual neurons respond to multilingual inputs. The authors identify and categorize language-related neurons into language-exclusive neurons (activated only for specific languages) and language-shared neurons (activated across multiple languages). Through extensive analysis of 20 open-source models across different families and generations, they discover that as LLMs evolve, shared neurons not only increase in proportion but also grow disproportionately in functional importance, eventually forming a compact set of language-agnostic neurons that support abstract reasoning beyond linguistic boundaries.

The key contributions include: (1) empirical evidence showing that shared neurons evolve from superficial overlaps in early models to critical language-agnostic components in advanced models, with their deactivation causing orders of magnitude greater performance degradation than exclusive neurons; (2) the introduction of a language-agnostic score metric that quantifies this evolution; and (3) a practical application of these insights through neuron-targeted training strategies that adapt based on a model's developmental stage.

**Questions:**

Suggestions:
- Perform an analysis of the statistics of the different neurons (where they exist in the model, their activation statistics compared to random, etc).
- Analyze multilinguality in terms of parameters, training tokens, and FLOPs instead of release date (deep research can probably accumulate these figures for you)
- Compare your neuron training to a PeFT method with a similar computation budget.

**Ethical Concerns:**

["NO or VERY MINOR ethics concerns only"]

**Final Justification:**

The authors have given a detailed rebuttal, which have addressed several of my concerns. I am raising my score from 2 to 3.

**Limitations:**

Yes

**Quality:**

2

**Strengths And Weaknesses:**

Strengths
- Very comprehensive comparison across model families and sizes.
- This is one of only a few papers that leverages an observation about model internals to develop a practical method.

Weaknesses
- I don't understand the emphasis on release date as an interesting axis. Parameter count, total tokens trained, and FLOPs are all more interesting covariates.
- In Figure 3 the stated pearson R ≈ 0.94 seems optimistic given visual scatter
- There is no analysis in terms of what layer the neurons are in the model, or there average activation magnitude/sparsity. I could imagine Figure 4 is fully explained by random neurons simply having different statistics from those specifically filtered to have a large effect on the final activations.
- Figure 2 is unnatural as a line plot and the grouping by size rather than generation obscures the more interesting comparison (how much shared neurons change as a function of size).
- Inconsistencies with prior results: In "On the Biology of a Large Language Model", Lindsey et al., showed that larger models have a greater degree multilingual overlap (intuitive). In contrast, Figure 1, shows no trends with respect to model size (not intuitive). The paper is also missing a reference to this paper.C
- I don't think the training method has much practical utility, compared to LoRA or other PEFT methods, as it requires an expensive preprocessing stage to find the relevant neurons. Comparing to LoRA is an easy baseline that is missing.

---

> ### Author Rebuttal · Authors · 2025-07-31
>
> # Response to Reviewer PaXX
>
> We appreciate your comments. Below we provide the point-to-point responses to address your concerns and clarify the misunderstandings of our proposed method. If you have additional questions, we would be pleased to discuss them with you.
>
> > **Comment 1:  Confusion of axis.** — “  I don't understand the emphasis on release date as an interesting axis…”
>
> Thank you for the thoughtful comment. Our primary focus is to investigate how the proportion and importance of shared neurons evolve with improving multilingual capabilities. In this context, the central axis of interest is a model’s *multilingual ability*. We introduced the release date as a proxy to provide an intuitive and chronological perspective on this evolution—particularly since, within the same model family (i.e., architecture and size), newer models tend to exhibit stronger multilingual performance due to improvements in training data quality, scale, or objective functions.
>
> We fully agree that parameter count and FLOPs are important covariates. In fact, parameter count is annotated in Figures 3 and 5. Since FLOPs are approximately proportional to parameter count for models of the same architecture and training length, we chose not to repeat this information. Figure 1 is intended to be a simplified summary of Figures 3 and 5, designed to visually convey the high-level trend in a concise and accessible way—hence the use of release date as the x-axis.
>
> We initially considered *training tokens* as a covariate, but found them often underreported or ambiguous (e.g., LLaMA series). In some cases (e.g., Qwen-1.5 series), models of different sizes share the same token count, making it hard to isolate the effect of training tokens from model size and release time.
>
> ---
> > **Comment 2:  Confusion of R.** — “In Figure 3 the stated pearson R ≈ 0.94 seems optimistic given visual scatter …”
>
> In Figure 3 and Figure 5, our primary goal is to examine the relationship between shared neuron ratio and multilingual ability **within the same model family and at comparable parameter scales**, as we believe that **models across different sizes and families are not directly comparable**. The reported Pearson and Spearman coefficients are the averages computed across these individual model groups.
>
> Although our focus is not on global analysis, we nonetheless report both the global correlation and the model familiy-level group-average correlation coefficients for completeness:
>
> - **Figure 3**
>
>   - Global Pearson: 0.427 (*p* = 0.060), Spearman: 0.409 (*p* = 0.073)
>   - Group-average Pearson: 0.591, Spearman: 0.494
>
> - **Figure 5**
>   - Global Pearson: 0.279 (*p* = 0.234), Spearman: 0.226 (*p* = 0.339)
>   - Group-average Pearson: 0.312, Spearman: 0.270
>
> We can see that, **even though our primary focus is not on global or family-level average correlations, there still exist correlations**, particularly for the **shared neuron proportion**, which partially supports the general trend we aim to highlight.
>
>
> ---
> > **Comment 3:  Lack of neuron analysis.** — “There is no analysis in terms of what layer the neurons are in the model …”
>
> Thank you for this insightful suggestion. We agree that analyzing the distribution of language-specific neurons across layers and components can provide further insight. To address the question of where language-shared and language-exclusive neurons exist in the model, we conducted a detailed analysis on LLaMA-3.1-8B.
>
> **Layerwise Distribution**:
>
> We calculate the proportion of **shared neurons** (common to all languages) and **exclusive neurons** (specific to one language) across layers:
>
> | Layer Range | Avg Shared (%) | Avg Exclusive (%) |
> | - | - | - |
> | 0–5         | 3.07           | 4.40              |
> | 6–25        | 3.11           | 2.89              |
> | 26–31       | 2.83           | 3.83              |
>
> The **layerwise distribution** highlights distinct allocation patterns for shared and exclusive neurons. We observe that **exclusive neurons** are more concentrated in the **early** and **late** stages of the model, whereas their proportion is relatively lower in the **middle layers**. This suggests that the model tends to encode **language-specific features** near the input and output boundaries.
>
> In contrast, **shared neurons** maintain a relatively stable presence across all layers. This indicates that shared neurons are more involved in processing **generalizable, cross-lingual representations**, and serve as a backbone for multilingual understanding throughout the network.
>
> **Component-level Distribution**:
>
> | Component | Shared (%) | Exclusive (%) (avg over languages) |
> | - | - | - |
> | QK        | 92.50      | 59.48                              |
> | VO        | 2.90       | 23.80                              |
> | FFN       | 4.59       | 16.71                              |
>
> Shared neurons are mainly found in **QK**, aligning with general attention patterns. Exclusive neurons are relatively more prominent in **VO** and **FFN**, indicating their more important role in language-specific output transformations. This distribution indicates that **shared and exclusive neurons may serve different roles, although they both belong to language-specific neurons.**
>
> **Illustration of random neurons**:
>
> Regarding the random neurons in Figure 4, we carefully designed the selection process to make a fair comparison. Specifically, we sampled two sets of random neurons—each matching the total number of shared and exclusive neurons respectively. These random neurons were uniformly sampled across all layers and components of the model under the assumption of a homogeneous distribution. We then selected the group that had a higher influence on model performance and reported this in Figure 4. This same group was also used in the ablation experiments as “random neurons” reported in Table 1. It is indeed expected that the distribution and activation behavior of random neurons may differ from those of the language-specific neurons we detected.
>
>
> ---
> > **Comment 4:  Size as axis.** — “Figure 2 is unnatural as a line plot and the grouping by size rather than generation…”
>
> Thank you for your insightful suggestion. Figure 1 presents a simplified analysis of models within the same generation but of different sizes. Specifically, we observe that larger models tend to have a lower proportion and reduced importance of shared neurons. We attribute this to increased parameter specialization in larger models [1], which naturally leads to fewer shared neurons.
>
> Although our current work primarily focuses on the evolution of abstract thought over time rather than across model sizes, we appreciate this valuable observation and will incorporate it into the revised version of the paper.
>
>
> ---
> > **Comment 5:  Inconsistent results.** — “Inconsistencies with prior results: In On the Biology of a Large Language Model…”
>
> Thank you for pointing this out. We acknowledge the omission of the reference and will include it in the revised version.
>
> We also wish to clarify a key methodological difference. Lindsey et al. analyze multilingual generalization from the **feature space**—focusing on hidden state reuse across languages. In contrast, our study operates in the **parameter space**, identifying **shared neurons** that are functionally important across languages and fixed in model structure.
>
> Due to these fundamental differences in approach—feature-level versus parameter-level analysis—the two works may lead to different empirical trends. For instance, our Figure 1 does not show a monotonic increase in shared neuron ratio with model size, which may reflect that **parameter sharing** across languages does not always scale proportionally with model capacity.
>
> We thank the reviewer again for highlighting this, which allowed us to more clearly distinguish our methodological contribution.
>
>
> ---
> > **Comment 6:  Additional baseline.** — “I don't think the training method has much practical utility, compared to LoRA or other PEFT methods…”
>
> We clarify that our training-stage intervention is not proposed as a new fine-tuning method, but rather as a way to validate our core finding: multilingual capability in LLMs can be enhanced by understanding and adjusting neuron-level contributions.
>
> While LoRA and other PEFT methods are widely adopted, they introduce additional parameters and adapter layers. In contrast, our method fine-tunes a small subset of existing shared neurons, aiming to produce a **fully multilingual model** without any added parameters. This reflects a different trade-off—targeted internal adjustment vs. parameter-efficient extension.
>
> Following your suggestion, we conducted LoRA experiments on Llama3.2-3B under similar parameter budgets. LoRA improved performance in some MGSM cases (e.g., German, Swahili), but degraded it in others, especially on MMMLU. Moreover, LoRA increased training time (2.2h on 2×H200, rank=48) compared to our method (1.5h), highlighting our method’s simplicity and efficiency without modifying the model architecture.
>
> |                   | MGSM   |        |        |        |        |               | MMMLU  |        |        |        |        |               |
> | - | - | - | - | - | - | - | - | - | - | - | - | - |
> |                   | zh     | de     | fr     | th     | sw     | average delta | zh     | de     | fr     | th     | sw     | average delta |
> | Llama-3.2-3B      | 40.80% | 57.20% | 42.40% | 35.20% | 30.80% | -             | 45.20% | 47.10% | 49.00% | 40.60% | 34.10% | -             |
> | Llama-3.2-3B-LoRA | 38.80% | 68.80% | 44.00% | 31.20% | 37.20% | 2.72%         | 44.40% | 45.79% | 47.74% | 39.58% | 32.50% | -1.12%        |
>
> [1] The Rise of Parameter Specialization for Knowledge Storage in Large Language Models. Arxiv 2025

---

> > ### Comment · Reviewer_PaXX · 2025-08-03
> >
> > I appreciate the authors' thoughtful response and additional experiments.
> >
> > I still think the FLOPs (using the 6 × N × D approximation) would be a more informative and principled covariate than release date.
> >
> > > Shared neurons are mainly found in QK
> >
> > This implies the analysis has been conducted over vector spaces without a privileged basis (i.e, the output of a linear transformation, such as QK, OV, and the residual stream) and not just the actual nonlinearities in the MLP layer. This is not an especially principled thing to do (see https://transformer-circuits.pub/2021/framework/index.html) and makes the results less compelling.

---

> > > ### Author Response · Authors · 2025-08-04
> > >
> > > > **Comment 1: FLOPs as axis.** — “ I still think the FLOPs. …”
> > >
> > > Thank you for the follow-up. We would like to clarify that our primary axis of interest is not the *release date*, but rather the *multilingual ability* of a model. The release date was used in Figure 1 merely as a proxy to reflect the chronological progression of multilingual performance *within the same model family*. We agree that FLOPs offer a more informative and principled perspective for comparing models across architectures or training regimes.
> > >
> > > If you are referring to **inference-time FLOPs**, we note that the approximation `6 × N × D` reduces to being proportional to `N` in our case, since all models are evaluated on the same test corpus with fixed input length `D`. As such, inference-time FLOPs correlate closely with parameter count, which we already provide in Figures 3 and 5. In fact, Figure 1 also visualizes how the shared neuron proportion and importance vary with parameter size `N` within each generation, and we discuss these trends in Comment 4.
> > >
> > > If you are referring to **training-time FLOPs**, where `D` corresponds to the number of pretraining tokens, we have now updated our analysis and include the results below. We define the *FLOPs score* as `6 × N × D`, with `N` in billions and `D` in trillions of tokens (i.e., FLOPs in units of T):
> > >
> > > | model_name   | shared_neuron_proportion | language_agnostic_score | FLOPs_score | multilingual_ability |
> > > |:-|--:|--:|--:|--:|
> > > | Qwen1.5-0.5B | 1.13421 | 3.41584 | 7.2 | 0.261990 |
> > > | Qwen2-0.5B | 1.83646 | 6.98906 | 36 | 0.337755 |
> > > | Qwen2.5-0.5B | 1.84428 | 9.19561 | 54 | 0.373706 |
> > > | Qwen1.5-1.8B | 0.92514 | 0.220784 | 23.76 | 0.321091 |
> > > | Qwen2-1.5B | 1.60181 | 5.22639 | 63 | 0.466117 |
> > > | Qwen2.5-1.5B | 1.59369 | 10.1366 | 162 | 0.521075 |
> > > | Qwen1.5-4B | 1.01834 | 6.08787 | 57.6 | 0.428578 |
> > > | Qwen2.5-3B | 1.84485 | 8.03691 | 324 | 0.580739 |
> > > | Llama-1-7B | 0.53719 | 0.770552 | - | 0.272332 |
> > > | Llama-2-7B | 2.55689 | 6.34957 | 84 | 0.303358 |
> > > | Llama-3-8B | 2.48528 | 1.64434 | 720 | 0.543645 |
> > > | Llama-3.1-8B | 2.56425 | 3.08567 | 720 | 0.548334 |
> > > | Qwen1.5-7B | 0.5625 | 0.010263 | 100.8 | 0.521890 |
> > > | Qwen2-7B | 1.49071 | 3.85643 | 294 | 0.634708 |
> > > | Qwen2.5-7B | 1.71733 | 5.96051 | 756 | 0.667487 |
> > > | Gemma-3-4B | 2.57932 | 6.38409 | 96 | 0.503646 |
> > > | Gemma-7B | 2.47807 | 5.96574 | 252 | 0.556681 |
> > > | Gemma-2-9B | 3.26421 | 8.03246 | 432 | 0.656689 |
> > > | Llama-3.2-1B | 1.55099 | 2.53049 | 54 | 0.269429 |
> > > | Llama-3.2-3B | 1.95304 | 4.40729 | 162 | 0.438506 |
> > >
> > >
> > > As discussed in our main paper, we observe that **within the same model family and parameter size**, newer generations tend to exhibit stronger multilingual ability, along with higher shared neuron proportion and importance. The increase in FLOPs may partially explain this evolution. However, we do not observe a consistent cross-family pattern between FLOPs and multilingual performance. This suggests that FLOPs alone are insufficient to account for multilingual generalization, and factors such as training objectives, data diversity, and architectural refinements also play crucial roles.
> > >
> > > > **Comment 2: Unprincipled analysis.** — “This implies the analysis has been conducted over vector spaces …”
> > >
> > > Thank you for raising this point. We would like to clarify a key distinction regarding the nature of our analysis. Contrary to the interpretation implied by the comment, **our analysis is not conducted in the activation or vector space**, but rather in the **parameter space**, specifically on the weight matrices of the model.
> > >
> > > As noted in the cited article ([Elhage et al., 2021], the concept of a "privileged basis" applies to **activations** (i.e., model states), not **parameters**. Indeed, the article explicitly states that the *query, key, and value weights of an attention head are parameters*, and thus, do not fall under the category of vector spaces where rotation invariance or privileged basis considerations apply. Therefore, the critique based on coordinate instability is **not applicable to our setting**.
> > >
> > > Furthermore, we would like to highlight that since the publication of [Elhage et al., 2021], **many subsequent studies have conducted insightful and rigorous analyses directly on QKV and MLP parameters [1-5]**, demonstrating both empirical validity and community acceptance of such approaches. Our work is aligned with and inspired by this growing body of research.
> > >
> > > We hope this clarifies the scope and theoretical grounding of our analysis, and we are open to any further discussion.
> > >
> > > [1] *Finding the Pillars of Strength for Multi-Head Attention*. ACL 2023
> > > [2] *Language-Specific Neurons: The Key to Multilingual Capabilities in Large Language Models*. ACL 2024
> > > [3] *Investigating Pattern Neurons in Urban Time Series Forecasting*. ICLR 2025
> > > [4] *Multilingual Knowledge Editing with Language-Agnostic Factual Neurons*. COLING 2025
> > > [5] *How Do Large Language Models Handle Multilingualism?* NeurIPS 2024

---

> > > > ### Author Response · Authors · 2025-08-07
> > > >
> > > > Thank you for your time and efforts for reviewing our paper.
> > > >
> > > > We noticed that you have submitted the final justification, but it is not visible to us. We would like to kindly confirm whether our response has addressed your concerns. If you have any further concerns, we would be happy to address them.

---

### Note · Authors · 2025-08-15

**Dear Reviewers and ACs,**

We are delighted that the most of the reviewers recognized the **novelty** of our work (Reviewers PaXX, AGNp, ysL3). We investigate an **important research question**—*how LLMs think*—and obtained **interesting and significant observations** that LLMs evolve a **compact set of language-agnostic neurons** supporting abstract reasoning beyond linguistic boundaries (Reviewers AGNp, oHcr). This finding is supported by a **comprehensive comparison across model families and sizes** (Reviewer PaXX). Building on this observation, we propose a **neuron-specific tuning method** to enhance the multilingual ability of LLMs, which is considered **highly original** (Reviewer ysL3). The paper is also noted to be **well-written and easy to follow** (Reviewers oHcr, ysL3) and recognized as **one of the few works leveraging internal model observations to develop a practical method** (Reviewer PaXX).

The reviewers also raised insightful and constructive concerns, and in response, we have made the following major revisions:

- **Analysis:** We analyzed the layerwise and component-level distribution of language-shared, language-exclusive, and language-unique neurons, providing new insights into how LLMs handle multilingual queries.
- **Clarifications:** We clarified how the threshold for selecting language-specific neurons was determined, how models were chosen in our training experiments, and the details of our correlation computation—an essential part of our work.
- **Additional Experiments:** We conducted triple-repetition experiments and reported error bounds. We also evaluated the cross-lingual ability of LLMs trained on a single language. Furthermore, we introduced two new LLM backbones (Qwen2.5-1.5B and Gemma2-9B), demonstrating the generality and effectiveness of our training method.
- **Baselines:** We compared our training method with LoRA, verifying the lightweight and effective nature of our approach.

Overall, we are **encouraged by the positive feedback** from most reviewers, noting that Reviewers AGNp and oHcr have raised their scores and Reviewer ysL3 expressed being *very satisfied* with our responses. We hope our work offers a **fresh perspective**, inspiring further exploration of **neuron-level analysis** in enhancing the **multilingual ability** of LLMs.


Thank you once again for your invaluable suggestions, support, and organization.

**Best regards,**
The Authors

---

### Decision · Program_Chairs · 2025-09-17

**Decision:**

Accept (poster)

**Comment:**

This papers investigated an important question on how LLMs think by analyzing how individual neurons respond to multi-lingual inputs. Based on the findings and observation of language-specific and language-agnostic neurons, the authors further proposed neuron-specific tuning method to improve the multilingual ability of LLMs. Overall, most reviewers are positive about the novelty of the work, the observed insights, and the extensive evaluations. The authors did a good job on addressing reviewers questions with additional empirical evidence, analysis, and clarifications. Although there are still some remaining concerns on the notion of agnostic score and discussions consistency (reviewer AGNp), it is expected that the authors can address them to further improve the clarity of the revised manuscript. As a result, an acceptance is recommended.